# DELTA: AN ONLINE DOCUMENT-LEVEL TRANSLATION AGENT BASED ON MULTI-LEVEL MEMORY

**Yutong Wang**[1][*] **Jiali Zeng**[2] **Xuebo Liu**[1][†] **Derek F. Wong**[3] **Fandong Meng**[2] **Jie Zhou**[2] **Min Zhang**[1]

[1]Institute of Computing and Intelligence, Harbin Institute of Technology, Shenzhen, China
[2]Pattern Recognition Center, WeChat AI, Tencent Inc, China
[3]NLP[2]CT Lab, Department of Computer and Information Science, University of Macau
`wangyutong@stu.hit.edu.cn`, `{liuxuebo,zhangmin2021}@hit.edu.cn`
`{lemonzeng,fandongmeng,withtomzhou}@tencent.com, derekfw@um.edu.com`

## ABSTRACT

Large language models (LLMs) have achieved reasonable quality improvements in machine translation (MT). However, most current research on MT-LLMs still faces significant challenges in maintaining translation consistency and accuracy when processing entire documents. In this paper, we introduce DELTA, a **D**ocument-lev**EL** **T**ranslation **A**gent designed to overcome these limitations. DELTA features a multi-level memory structure that stores information across various granularities and spans, including Proper Noun Records, Bilingual Summary, Long-Term Memory, and Short-Term Memory, which are continuously retrieved and updated by auxiliary LLM-based components. Experimental results indicate that DELTA significantly outperforms strong baselines in terms of translation consistency and quality across four open/closed-source LLMs and two representative document translation datasets, achieving an increase in consistency scores by up to 4.58 percentage points and in COMET scores by up to 3.16 points on average. DELTA employs a sentence-by-sentence translation strategy, ensuring no sentence omissions and offering a memory-efficient solution compared to the mainstream method. Furthermore, DELTA improves pronoun and context-dependent translation accuracy, and the summary component of the agent also shows promise as a tool for query-based summarization tasks. The code and data of our approach are released at `https://github.com/YutongWang1216/DocMTAgent`.

## 1 INTRODUCTION

Large language models (LLMs) such as GPT-4 (OpenAI, 2023) have recently demonstrated reasonable performance on the machine translation (MT) task within the natural language processing domain (Garcia & Firat, 2022; Hendy et al., 2023; Zhang et al., 2023; Siu, 2023; Jiao et al., 2023). Numerous studies have been carried out to further unleash LLMs' potential for MT (Ghazvininejad et al., 2023; Peng et al., 2023; Zeng et al., 2024; He et al., 2024; Sun et al., 2024; Wang et al., 2024c; Liu et al., 2024). However, the majority of these researches mainly focus on sentence-level translation, operating under the strong assumption that source sentences are independent of one another. This isolated approach may fail to model the discourse structure and overlook the coherence in continuous document texts (Scarton & Specia, 2015; Bawden et al., 2018).

Document-level machine translation (DocMT) systems have been receiving growing focus in recent years, which involves the whole document or some part of it to capture more context information to guide the translation process (Kim et al., 2019; Maruf et al., 2021). Researchers find that modeling discourse phenomena (Bawden et al., 2018) while translating the whole document helps increase the coherence and consistency in the generated translation (Maruf & Haffari, 2018; Wang et al., 2017; Zhang et al., 2018; Tan et al., 2019). Recently, a few studies have proposed to introduce LLMs to the DocMT task, utilizing their inherent context information modeling and long text processing capabilities (Wang et al., 2023b; Wu & Hu, 2023; Wu et al., 2024a). However, existing DocMT-LLMs

---

[*] Work was done when Yutong Wang was interning at Pattern Recognition Center, WeChat AI, Tencent Inc.
[†] Xuebo Liu is the corresponding author.

still suffer from critical issues such as occasional content omissions and terminology translation inconsistency (Karpinska & Iyyer, 2023). These issues seriously affect the reliability of the developed system, especially when accurate document translations are required.

LLM-based autonomous agents equipped with specially designed memory components can efficiently store and retrieve key information embedded in the environment. These data assist the inference process of LLMs, facilitating the handling of complex tasks and environments through self-directed planning and actions (Wang et al., 2024a; 2023a; Park et al., 2023; Lee et al., 2024). Inspired by this, we propose DELTA, an online Document-levEL Translation Agent based on multi-level memory components. Specifically, we store information in four memory components: Proper Noun Records, Bilingual Summary, Long-Term Memory, and Short-term Memory, and utilize LLMs to update and retrieve them. Proper Noun Records maintain a repository of previously encountered proper nouns and their initial translations within the document, ensuring consistency by reusing the same translation for each subsequent occurrence of the same proper noun. The Bilingual Summary contains summaries of both the source and target texts, capturing the core meanings and genre characteristics of the documents to enhance translation coherence. Long-Term Memory and Short-Term Memory store contextual sentences over broader and narrower scopes, respectively. Long-Term Memory is accessed by LLMs to retrieve sentences most relevant to the current source sentence, while Short-Term Memory provides instant context to support the translation process. During translation, sentence pairs are drawn from the Long-Term and Short-Term memory as the exemplars for few-shot learning demonstration, and the proper noun translation records and bilingual summaries are also integrated into the prompt for DocMT-LLMs as auxiliary information.

Experimental results indicate that DELTA achieves improvements in both translation consistency and quality. For translation consistency, DELTA achieves an average improvement of 4.36 percentage points across four translation directions from English and 4.58 percentage points across four directions into English. In terms of translation quality, DELTA yields an average improvement of 3.14 COMET points for four translation directions from English and 3.16 COMET points for four directions into English. Moreover, DELTA translate documents in a sentence-by-sentence manner (following an online approach) to avoid content omissions, ensuring sentence-level alignment of target documents with source documents. This manner also prevents memory bloat caused by data accumulation, making it more suitable for practical application scenarios.

Our main contributions are summarized as follows:

- We develop DELTA, an online DocMT agent employing a multi-level memory structure, which stores information across different granularities and spans.

- We demonstrate that DELTA substantially improves the consistency and quality of document translations. Additionally, the summary component of DELTA can function as an independent tool for query-based summarization tasks.

- We certificate that the sentence-wise translation approach employed by DELTA incurs a lower memory cost compared to existing document translation methods.

- We observe that DELTA is particularly effective in maintaining translation consistency over expended spans. Moreover, it enhances the pronoun translation accuracy in the document.

## 2 RELATED WORK

**Document-Level Machine Translation** In recent years, studies on DocMT have achieved rich results (Kim et al., 2019; Maruf et al., 2021; Fernandes et al., 2021). These studies can be separated into two categories. Studies of the first group employ a document-to-sentence (Doc2Sent) approach, where the source-side context sentences are encoded to generate the current target sentence (Wang et al., 2017; Tan et al., 2021; Lyu et al., 2021). However, these approaches suffer from limitations caused by separated encoding modules of the current sentences and their context (Sun et al., 2022; Bao et al., 2021), as well as the failure to utilize target-side context (Li et al., 2023b). Studies of the second group employ a document-to-document (Doc2Doc) approach, where the translation unit is extended from a single sentence to multiple sentences (Zhang et al., 2020; Liu et al., 2020; Lupo et al., 2022; Bao et al., 2021; Li et al., 2023b).

| Window | LTCR-1 | LTCR-1$_f$ | #Missing Sents | sCOMET | dCOMET |
|--------|--------|-----------|----------------|--------|--------|
| 1 | 75.09 | 88.24 | **0** | 84.04 | 6.62 |
| 5 | 80.49 | 88.15 | **0** | **84.30** | **6.70** |
| 10 | 79.65 | 90.81 | 2 | 84.27 | 6.65 |
| 30 | 83.08 | 95.83 | 8 | 83.88 | 6.69 |
| 50 | **86.94** | **95.90** | 10 | 83.70 | 6.66 |

Table 1: Translation results with different translation window sizes. "#Missing Sents" represents the number of missing target sentences in the translated document.

**Autonomous Agents**  LLM-based autonomous agents have recently achieved remarkable performance in various NLP tasks. Park et al. (2023); Wang et al. (2023a); Lee et al. (2024) deal with long-context understanding and processing tasks by introducing carefully designed memory and retrieval workflows. Xu et al. (2024); Wang et al. (2024c); Feng et al. (2024) prompt LLMs to evaluate their own outputs and conduct refinement accordingly to improve the quality of the outputs. Li et al. (2023a); Liang et al. (2024); Li et al. (2024); Wu et al. (2024b) enhance the performance of LLMs on specific tasks by multi-agent interaction.

Our method in this paper represents a Doc2Sent approach implemented through an LLM-based automatic agent. Instead of simply encoding source-side context to generate target sentences, our method directs LLMs to retrieve key information across varying granularities and spans, and store this data in memory components. During document translation, relevant information is incorporated into the prompts for DocMT-LLMs to assist the translation process.

## 3 MOTIVATION

### 3.1 MAIN CHALLENGES FOR DOCMT-LLMS

Due to the maximum context limitation inherent in LLMs, translating a lengthy document in a single pass becomes unfeasible. A conventional strategy involves segmenting the document into smaller translation windows and translating them sequentially. In our study, we initially leverage the `GPT-3.5-turbo-0125` model to translate the IWSLT2017 En $\Rightarrow$ Zh test set, comprising 12 documents sourced from TED talks. We employ a window of size $l$ to facilitate document translation, where $l$ source sentences are simultaneously processed to generate $l$ hypothesis sentences. Once all source sentences are translated, they are concatenated to form the complete target document. The primary challenges associated with DocMT-LLMs arise from the following two aspects.

**Translation Inconsistency**  Given a source document $\boldsymbol{D}_s = (s_1, s_2, \ldots, s_N)$ and its corresponding target document $\boldsymbol{D}_t = (t_1, t_2, \ldots, t_N)$, if there exists a proper noun $p \in P$ ($P$ denotes the set of all proper nouns in $\boldsymbol{D}_s$, including names of people, locations, and organizations), and $p$ appears multiple times in $\boldsymbol{D}_s$, we expect that all occurrences of its translation in $\boldsymbol{D}_t$ should be consistent.

Lyu et al. (2021) propose the Lexical Translation Consistency Ratio (LTCR), a metric that quantifies the proportion of consistent translation pairs among all proper noun translation pairs in the target document. However, we argue that the translations of the proper nouns are supposed to not only maintain consistency throughout the document but also align their first appearance. This consideration is particularly important for enhancing the reading experience of audiences. Therefore, we introduce the **LTCR-1** metric for the DocMT-LLMs, which calculates the proportion of proper noun translations that are consistent with the initial translation within the document:

$$\text{LTCR-1}(\boldsymbol{D}_s, \boldsymbol{D}_t) = \frac{\sum_{p \in P} \sum_{i=2}^{k_p} \mathbb{1}(\mathcal{T}_i(p) = \mathcal{T}_1(p))}{\sum_{p \in P} (k_p - 1)} \tag{1}$$

$\mathcal{T}_i(p)$ represents the $i$-th translation of $p$ in $\boldsymbol{D}_t$, and $k_p$ denotes the number of occurrences of $p$ in the document. The indicator function $\mathbb{1}(\mathcal{T}_i(p) = \mathcal{T}_1(p))$ returns 1 if the translations $\mathcal{T}_i(p)$ and $\mathcal{T}_1(p)$ are identical, and 0 otherwise. The numerator is the number of times the proper nouns appear again and their translation remains the same as their first appearance, and the denominator represents the sum of all occurrences except the first one of all proper nouns. To compute this metric, we

initially annotate all proper nouns in the source document using spaCy[1]. Subsequently, we utilize the token align tool awesome-align (Dou & Neubig, 2021)[2] to determine the translations of these proper nouns in the target document. To mitigate the impact of errors from the alignment tool, we introduce a fuzzy match version of this metric, where two proper noun translations are considered consistent when one is a substring of the other:

$$\text{LTCR-1}_\text{f}(\boldsymbol{D}_s, \boldsymbol{D}_t) = \frac{\sum_{p \in P} \sum_{i=2}^{k_p} \mathbb{1}(\mathcal{T}_i(p) \subseteq \mathcal{T}_1(p) \vee \mathcal{T}_1(p) \subseteq \mathcal{T}_i(p))}{\sum_{p \in P} (k_p - 1)} \tag{2}$$

As shown in Table 1, translating every sentence separately (window size = 1) causes poor translation consistency. An example is illustrated in Appendix A. Increasing the window size consistently leads to higher scores across all three consistency metrics. This suggests that when more sentences are processed within a single translation pass, the LLM is better able to model discourse phenomena and maintain consistent translation of proper nouns throughout the document. However, due to the inherent limitations in the context length of LLMs, resolving translation inconsistencies cannot be achieved solely by indefinitely expanding the window size.

**Translation Inaccuracy**    When employing a large window size for document translation, LLMs tend to process the input source sentences as cohesive documents rather than as individual sentences. As a result, the model prioritizes maintaining the general meaning of the text and loses track of the detailed information in each sentence. This can lead to undertranslation issues and a decline in translation quality (Karpinska & Iyyer, 2023; Wu et al., 2024a). We utilize two neural metrics to assess the quality of document translation. The first is the sentence-level COMET (sCOMET) score[3], for which we utilize the model Unbabel/wmt22-comet-da to obtain the scores. The second metric is the document-level COMET (dCOMET) score[4] proposed by Vernikos et al. (2022), for which we use wmt21-comet-qe-mqm[5] to derive reference-free scores. In calculating this document-level metric, the model encodes previous sentences as context rather than encoding only the hypothesis, making this approach more accurate for evaluating document translations.

As illustrated in Table 1, an increase in window size correlates with a higher tendency for the LLM to omit sentences from the source document, resulting in missing translations in the target document. An example of this undertranslation issue is presented in Appendix A. Additionally, quality metrics such as sCOMET and dCOMET do not demonstrate a consistent improvement with larger translation windows. Therefore, we conclude that translating documents in batches of several sentences at a time may introduce translation inaccuracy issues. These concerns are particularly significant in contexts where precise translations are essential, such as in technical manuals or official documents.

## 3.2    WHY USING A DOC2SENT APPROACH?

Previous experiments indicate that translating a document by processing multiple sentences at once may occasionally result in sentence omissions. Although human translators often translate entire paragraphs simultaneously, which can also lead to occasional omissions, their underlying translation mechanism is fundamentally distinct from that of DocMT-LLMs. DocMT-LLMs are prone to omitting source sentences due to hallucination issues or limited capabilities in handling long texts effectively. Therefore, we argue that, at this moment, a Doc2Sent approach offers a more promising alternative for DocMT-LLMs to produce precise and high-quality document translations. In our study, we provide LLMs with contextual information from the document while asking them to translate each source sentence separately. Once all sentences are translated, we concatenate the target sentences to form the final target document.

---

[1] https://spacy.io/

[2] https://github.com/neulab/awesome-align/

[3] https://github.com/Unbabel/COMET/

[4] https://github.com/amazon-science/doc-mt-metrics/

[5] https://unbabel-experimental-models.s3.amazonaws.com/comet/wmt21/wmt21-comet-qe-mqm.tar.gz

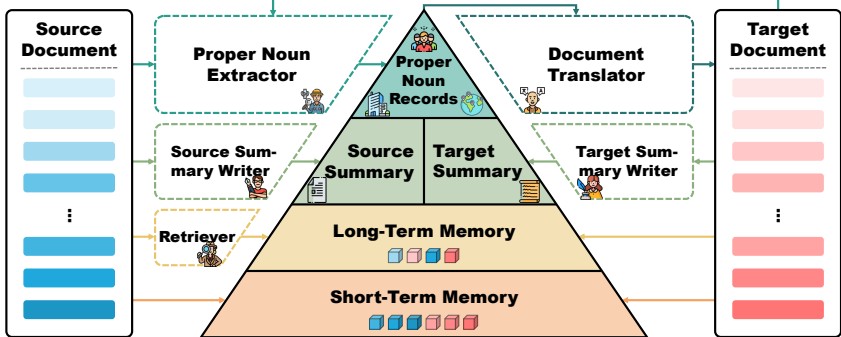

Figure 1: Framework of DELTA. The modules outlined with dashed lines represent the multi-level memory components, while those outlined with solid lines denote the LLM-based components. Memories closer to the top are more global, abstract, and densely packed with information. During translation, memory information is retrieved and incorporated into the translator LLM's prompt. After the translation of each sentence, the LLM-based components extract key information from both the source and target documents and update the multi-level memory components.

## 4 DELTA: DOCMT AGENT BASED ON MULTI-LEVEL MEMORY

Considering the multi-granularity and multi-scale of key information in the document during translation, we introduce **DELTA**, an online DocMT agent. DELTA employs a multi-level memory stream that captures and preserves critical information encountered throughout the translation process. This memory stream accommodates a wide range of perspectives, spanning from recent to historical, concrete to abstract, and coarse-grained to fine-grained details. DELTA translate the source document in a sentence-by-sentence manner while updating its memory in real-time. This approach addresses the context limitations of large language models and ensures the generation of a sentence-level aligned target document, thereby preserving both the quality and rigorousness of the translation. The main framework of DELTA is illustrated in Figure 1, the algorithm of DELTA is detailed in Algorithm 1, and the prompts used for each module are given in Appendix C.

**Proper Noun Records** The first level of the agent's memory component we introduce is a dictionary called the Proper Noun Records $\mathcal{R}$ to store proper nouns $p$ in the document along with their translations upon first encounter $\mathcal{T}_1(p)$ within the document: $\mathcal{R}^{(i)} = \{(p, \mathcal{T}_1(p)) \mid p \in s_j, 1 \leq j < i\}$, where $\mathcal{R}^{(i)}$ represents the state of $\mathcal{R}$ before translating the $i$-th sentence, and the same applies to other components. When translating the subsequent sentence $s_i$, the agent consults $\mathcal{R}^{(i)}$ to obtain all recorded proper nouns that are also contained in $s_i$: $\hat{\mathcal{R}}^{(i)} = \{(p, \mathcal{T}_1(p)) \mid p \in s_i, (p, \mathcal{T}_1(p)) \in \mathcal{R}^{(i)}\}$.

The Proper Noun Records are continuously updated by an LLM-based component known as the Proper Noun Extractor $\mathcal{L}_{\text{Extract}}$. After each sentence is translated, it extracts newly encountered proper nouns from the source sentence and their translations from the target sentence $\mathcal{L}_{\text{Extract}}(s_i, t_i) = \{(p, \mathcal{T}_j(p)) \mid p \in s_i, \mathcal{T}_j(p) \in t_i, \forall(p', \mathcal{T}_k(p)) \in \mathcal{R}^{(i)}, p \neq p'\}$ and add them to $\mathcal{R}$.

**Bilingual Summary** Unlike previous studies (Wang et al., 2023a; Lee et al., 2024), our research implements a bilingual summary approach as the second level of the agent's memory component to address the challenges of extensive context on both the source and target sides. We maintain a pair of summaries throughout the translation process to enhance accuracy and fluency. The Source-Side Summary $\mathcal{A}_s$ encapsulates the main content, domain, style, and tone of the previously translated sections of the document. This summary serves to preserve a coherent understanding of the text's overall context, thereby aiding the LLMs in producing more accurate translations. Conversely, the Target-Side Summary $\mathcal{A}_t$ focuses solely on the main content of the previously translated target text.

The pair of summaries are generated by two LLM-based components of the agent: the Source Summary Writer $\mathcal{L}_{\text{WriteS}}$ and the Target Summary Writer $\mathcal{L}_{\text{WriteT}}$. These summaries are updated every $m$ sentences through a two-step process. Initially, the writers generate segment summaries for the last $m$ sentences from both the source and target texts: $\tilde{\mathcal{A}}_s^{(i+1)} = \mathcal{L}_{\text{WriteS}}(s_{i-m+1}, \ldots, s_i)$,

---

**Algorithm 1:** The Overall Framework of DELTA

---

**input** : Source document $\boldsymbol{D}_s = \{s_1, \ldots, s_N\}$, Large Language Model $\mathcal{L}$, Proper Noun
        Records $\mathcal{R} = \emptyset$, Source-Side Summary $\mathcal{A}_s = \emptyset$, Target-Side Summary $\mathcal{A}_t = \emptyset$,
        Short-Term Memory $\mathcal{M} = \emptyset$, Long-Term Memory $\mathcal{N} = \emptyset$
**output:** Target document $\boldsymbol{D}_t = \{t_1, \ldots, t_N\}$
$\boldsymbol{D}_t \leftarrow \emptyset$
**for** $i = 1$ *to* $N$ **do**
  | /* Retrieve memory                                                                 */
  | $\hat{\mathcal{R}} \leftarrow \{(p, \mathcal{T}_1(p)) \mid p \in s_i, (p, \mathcal{T}_1(p)) \in \mathcal{R}\}$       /* Search Proper Noun Records */
  | $\hat{\mathcal{N}} \leftarrow \mathcal{L}_{\text{Retrieve}}(s_i, \mathcal{N})$   /* Match $n$ relative sentences from Long-Term Memory */
  | /* Translate with hybrid memory information                                    */
  | $t_i \leftarrow \mathcal{L}_{\text{Translate}}(s_i, \hat{\mathcal{R}}, \hat{\mathcal{N}}, \mathcal{A}_s, \mathcal{A}_t, \mathcal{M})$
  | $\boldsymbol{D}_t \leftarrow \boldsymbol{D}_t \cup \{t_i\}$                 /* Add hypothesis to target document */
  | /* Update memory                                                                 */
  | $\mathcal{R} \leftarrow \mathcal{R} \cup \mathcal{L}_{\text{Extract}}(s_i, t_i)$   /* Extract new proper nouns and add to records */
  | $\mathcal{N} \leftarrow \mathcal{N}[-l + 1:] \cup \{(s_i, t_i)\}$      /* Last $l$ sentences as Long-Term Memory */
  | $\mathcal{M} \leftarrow \mathcal{M}[-k + 1:] \cup \{(s_i, t_i)\}$    /* Last $k$ sentences as Short-Term Memory */
  | **if** $i \mod m = 0$          /* Update Bilingual Summary every $m$ sentences */
  | **then**
  |   | /* Generate source and target segment summaries                      */
  |   | $\tilde{\mathcal{A}}_s \leftarrow \mathcal{L}_{\text{WriteS}}(s_{i-m+1}, \ldots, s_i)$  $\tilde{\mathcal{A}}_t \leftarrow \mathcal{L}_{\text{WriteT}}(t_{i-m+1}, \ldots, t_i)$
  |   | /* Merge segment summaries into document summaries              */
  |   | $\mathcal{A}_s \leftarrow \mathcal{L}_{\text{MergeS}}(\mathcal{A}_s, \tilde{\mathcal{A}}_s)$         $\mathcal{A}_t \leftarrow \mathcal{L}_{\text{MergeT}}(\mathcal{A}_t, \tilde{\mathcal{A}}_t)$
  | **end**
**end**

---

$\tilde{\mathcal{A}}_t^{(i+1)} = \mathcal{L}_{\text{WriteT}}(t_{i-m+1}, \ldots, t_i)$. Subsequently, these segment summaries are merged with the previous overall summaries for both sides to summary mergers to obtain new overall summaries: $\mathcal{A}_s^{(i+1)} = \mathcal{L}_{\text{MergeS}}(\mathcal{A}_s^{(i)}, \tilde{\mathcal{A}}_s^{(i+1)})$, $\mathcal{A}_t^{(i+1)} = \mathcal{L}_{\text{MergeT}}(\mathcal{A}_t^{(i)}, \tilde{\mathcal{A}}_t^{(i+1)})$. This process is repeated iteratively until all sentences in the source document have been read.

**Long-Term & Short-Term Memory** The last two levels of the agent's memory component are the Long-Term Memory and the Short-Term Memory, respectively. These two components are designed to address the requisite coherence across document-level translations. The Short-Term Memory $\mathcal{M}$ retains the last $k$ source sentences along with their corresponding translations, where $k$ represents a relatively small number: $\mathcal{M}^{(i)} = \{(s_{i-k}, t_{i-k}), \ldots, (s_{i-1}, t_{i-1})\}$. This component is specifically designed to capture immediate contextual information in adjacent sentences, which is then seamlessly integrated into the translation prompt, serving as the context for the current sentence.

Similarly, the Long-Term Memory $\mathcal{N}$ component preserves a broader range of context by maintaining a window of the last $l$ sentences from the source document, with $l$ being significantly greater than $k$, storing extended coherent information throughout the document. Before translating a given source sentence, an LLM-based component called the Memory Retriever $\mathcal{L}_{\text{Retrieve}}$ chooses $n$ source sentences that are most relevant to the current translation query: $\hat{\mathcal{N}}^{(i)} = \mathcal{L}_{\text{Retrieve}}(s_i, N^{(i)})$. These $m$ sentences with their translations are subsequently employed as demonstration exemplars.

**Document Translator** We utilize an LLM-based component called Document Translator $\mathcal{L}_{\text{Translate}}$ to perform the final translation process. Information from the multi-level memory is integrated into the prompt to support the translator in producing high-quality and consistent translations: $t_i = \mathcal{L}_{\text{Translate}}(s_i, \hat{\mathcal{R}}^{(i)}, \hat{\mathcal{N}}^{(i)}, \mathcal{A}_s^{(i)}, \mathcal{A}_t^{(i)}, \mathcal{M}^{(i)})$. The sentence-by-sentence approach ensures that the resulting target document is consistently aligned with the source document at the sentence level, effectively minimizing the risk of missing target sentences. Additionally, this method allows for straightforward evaluation of translation quality using sentence-level metrics such as sCOMET.

## 5 EXPERIMENTS

### 5.1 SETTINGS

**Datasets & Metrics** We conduct our experiments on the two test sets. The first is the tst2017 test sets from the IWSLT2017 translation task[6] (Akiba et al., 2004), which consists of parallel documents sourced from TED talks, covering 12 language pairs. Our experiments are conducted on eight language pairs: En ⇔ Zh, De, Fr, and Ja. There are 10 to 12 sentence-level aligned parallel

---

[6]https://wit3.fbk.eu/2017-01-d/

| System | En ⇒ Xx | | | | Xx ⇒ En | | | |
|---|---|---|---|---|---|---|---|---|
| | sCOMET | dCOMET | LTCR-1 | LTCR-1$_f$ | sCOMET | dCOMET | LTCR-1 | LTCR-1$_f$ |
| NLLB | 82.11 | 6.36 | 74.56 | 81.87 | 84.10 | 6.98 | 79.03 | 90.76 |
| GOOGLE | 80.41 | 5.83 | 81.38 | 84.72 | 80.17 | 5.96 | 81.43 | 90.81 |
| | | | | GPT-3.5-Turbo | | | | |
| Sentence | 84.80 | 6.58 | 77.06 | 82.81 | 84.47 | 7.05 | 81.98 | 91.86 |
| Context | 85.40 | 6.70 | 77.34 | 83.12 | **84.97** | **7.15** | 85.03 | 95.27 |
| Doc2Doc | – | 6.62 | 79.12 | 86.39 | – | 6.96 | 85.17 | 92.98 |
| DELTA | **85.58** | **6.73** | **82.96** | **88.83** | 84.95 | **7.15** | **86.53** | **96.26** |
| | | | | GPT-4o-mini | | | | |
| Sentence | 81.51 | 6.35 | 78.59 | 85.07 | 84.01 | 6.99 | 81.42 | 91.34 |
| Context | 84.78 | 6.65 | 80.01 | **86.99** | 84.95 | 7.15 | 84.40 | 94.34 |
| Doc2Doc | – | 6.75 | 80.54 | 85.39 | – | 7.01 | 83.50 | 93.39 |
| DELTA | **85.85** | **6.80** | **81.80** | 86.33 | **85.26** | **7.24** | **85.25** | **95.89** |
| | | | | Qwen2-7B-Instruct | | | | |
| Sentence | 80.03 | 5.96 | 73.91 | 79.54 | 77.10 | 6.48 | 76.39 | 87.94 |
| Context | 80.84 | **6.08** | 79.59 | 85.35 | 83.09 | **6.84** | 81.48 | 92.56 |
| Doc2Doc | – | 5.83 | 77.32 | 84.59 | – | 6.59 | **85.03** | **93.68** |
| DELTA | **81.02** | 6.07 | **80.09** | **87.78** | **83.36** | **6.84** | 82.05 | 93.30 |
| | | | | Qwen2-72B-Instruct | | | | |
| Sentence | 78.53 | 5.97 | 79.54 | 85.09 | 80.53 | 6.73 | 82.25 | 92.05 |
| Context | 80.79 | 6.22 | 79.14 | 85.40 | 83.27 | 6.99 | 82.86 | 92.21 |
| Doc2Doc | – | 6.45 | 73.58 | 78.64 | – | 6.87 | 83.00 | 90.74 |
| DELTA | **84.99** | **6.66** | **81.66** | **88.34** | **85.19** | **7.21** | **86.53** | **96.48** |
| | | | | Average | | | | |
| Sentence | 81.22 | 6.21 | 77.27 | 83.13 | 81.53 | 6.81 | 80.51 | 90.80 |
| Context | 82.95 | 6.41 | 79.02 | 85.21 | 84.07 | 7.03 | 83.44 | 93.59 |
| Doc2Doc | – | 6.41 | 77.64 | 83.75 | – | 6.86 | 84.18 | 92.70 |
| DELTA | **84.36** | **6.57** | **81.63** | **87.82** | **84.69** | **7.11** | **85.09** | **95.48** |

Table 2: Test results on the IWSLT2017 dataset. Since the translations produced by the Doc2Doc method are not aligned at the sentence level with the source text, we do not report the sCOMET scores for this method. The highest score in each block is highlighted in **bold font** The results in the "Average" block represent the mean scores across the four backbone models.

documents with approximately 1.5K sentences for each language pair. The second is Guofeng Web-novel[7] (Wang et al., 2023c; 2024b), a high-quality and discourse-level corpus of web fiction. We conduct our experiments on the Guofeng V1 TEST_2 set in the Zh ⇒ En direction. The detailed dataset statistics are demonstrated in Appendix B. We employ LTCR-1 and LTCR-1$_f$ for proper noun translation consistency evaluation and adopt sCOMET and dCOMET as translation quality metrics, which are all introduced in §3.1.

**Models and Hyperparameters** In this work, we utilize two versions of GPT models, `GPT-3.5-Turbo-0125` and `GPT-4o-mini`, as our base models. We get access to these models through the official API provided by OpenAI[8]. We also introduce the open-source `Qwen2-7B-Instruct`[9] and `Qwen2-72B-Instruct`[10] in our experiments. The `max_new_tokens` is set to 2048 and other hyper-parameters remain default. The updating window of Bilingual summary $m$ and length of Long-Term Memory $l$ are set to 20. The number of retrieved relative sentences from Long-Term Memory $n$ is set to 2. The length of Short-Term Memory $k$ is set to 3.

**Baseline Methods** We include the following three approaches as our baselines: a) **Sentence**: We employ the same LLMs but conduct a sentence-level translation process to obtain the baseline results. b) **Context**: We follow Wu et al. (2024a) to provide the LLMs with three previously obtained source-target sentence pairs to integrate more contextual information and adapt LLMs for document-level translation tasks. c) **Doc2Doc**: We reproduce the approach proposed by Wang et al. (2023b), translating 10 sentences in a single conversation turn and processing the entire document within a

---

[7] https://github.com/longyuewangdcu/GuoFeng-Webnovel/

[8] https://platform.openai.com/docs/guides/text-generation/

[9] https://huggingface.co/Qwen/Qwen2-7B-Instruct/

[10] https://huggingface.co/Qwen/Qwen2-72B-Instruct/

| System | sCOMET | dCOMET | LTCR-1 | LTCR-1$_\text{f}$ | sCOMET | dCOMET | LTCR-1 | LTCR-1$_\text{f}$ |
|---|---|---|---|---|---|---|---|---|
| | `GPT-3.5-Turbo` | | | | `GPT-4o-mini` | | | |
| Sentence | 77.62 | 3.07 | 61.58 | 78.82 | 77.87 | 3.10 | 58.82 | 70.59 |
| Context | **78.57** | **3.19** | 70.10 | 81.37 | 78.56 | 3.19 | 64.32 | 74.37 |
| Doc2Doc | – | 2.82 | 77.46 | 89.02 | – | 2.96 | 82.04 | 91.62 |
| DELTA | 78.45 | 3.17 | **85.57** | **96.52** | **78.77** | **3.34** | **88.94** | **96.48** |
| | `Qwen2-7B-Instruct` | | | | `Qwen2-72B-Instruct` | | | |
| Sentence | 73.65 | 2.62 | 37.00 | 50.00 | 75.15 | 2.98 | 58.00 | 71.50 |
| Context | 76.54 | 3.01 | 52.82 | 61.54 | 77.87 | 3.20 | 58.21 | 70.15 |
| Doc2Doc | – | 2.69 | 73.25 | 84.08 | – | 2.77 | 80.79 | 90.07 |
| DELTA | **76.95** | **3.10** | **85.50** | **94.00** | **78.32** | **3.31** | **86.93** | **95.98** |

Table 3: Test results on the Guofeng dataset.

single chat box, thereby leveraging LLMs' long-term modeling ability. In computing the metric scores for the Doc2Doc results, we first perform sentence alignment using Bleualign[11] to obtain aligned source-target documents, then we calculate the involved metrics. Furthermore, we also introduce the results of `NLLB-3.3B` (Costa-jussà et al., 2022) and `GoogleTrans`[12] for comparison.

## 5.2 Results

**Test Results on IWSLT2017** The main experiment results on the IWSLT2017 test set are demonstrated in Table 2. For more detailed scores, please refer to Appendix D. It is evident that DELTA outperforms baseline approaches on LTCR-1 and LTCR-1$_\text{f}$ metric scores across nearly all translation directions and models. This indicates that our approach yields significant enhancements in proper noun translation consistency for document-level translation. Furthermore, DELTA significantly improves the overall quality of document translation, as evidenced by consistently higher sCOMET and dCOMET scores. The superior dCOMET scores indicate that DELTA effectively captures contextual information to support the translation process. Translation consistency is improved significantly in the En $\Rightarrow$ Zh direction, while gains in directions like En $\Rightarrow$ De are modest. For instance, with `GPT-3.5-Turbo`, LTCR-1 improves by 6.17 percentage points (86.44 vs. 80.27) for En $\Rightarrow$ Zh, but only 1.40 points (93.46 vs. 92.06) for En $\Rightarrow$ De (see Table 13 of Appendix D). This disparity stems from linguistic differences: English proper nouns require conversion into Chinese characters, posing challenges for maintaining consistency, whereas in German, they can be directly copied due to the shared alphabet. Despite this, a reasonable LTCR-1 improvement in En $\Rightarrow$ De still demonstrates our method's effectiveness. The p-values of t-tests for DELTA vs Sentence/Context in translation quality are less than 0.05 for En $\Leftrightarrow$ Xx, whereas that in translation consistency are less than 0.05 in En $\Leftrightarrow$ Zh. We also test DELTA on the low-resource language pair, as demonstrated in Appendix F.

**Test Results on Guofeng** The test results on Guofeng are illustrated in Table 3. Our approach achieves superior results across almost all metrics and backbone models, demonstrating its robustness to data of the novel domain. Notably, stronger models, such as `GPT-4o-mini` and `Qwen2-72B-Instruct`, achieve greater improvements in translation consistency and quality metrics. This suggests that the stronger the backbone models are, the more substantial the gains achieved by DELTA. The Guofeng test set poses particular challenges for maintaining translation consistency due to the prevalence of proper nouns (mainly names) in the source text. Nevertheless, DELTA demonstrates significant improvements in the relevant metrics, with an increase in LTCR-1 of up to 48.50 percentage points (85.50 vs. 37.00). This indicates that DELTA represents a strong tool for addressing translation inconsistency issues and holds great potential for novel translation, as it effectively reduces inconsistent noun translations, thereby minimizing potential confusion for readers.

## 6 Analysis

**Ablation Study** Table 4 presents an ablation study in the En $\Rightarrow$ Zh direction using `GPT-3.5-Turbo-0125` as the backbone model. For more detailed ablation studies of each memory

---

[11]`https://github.com/rsennrich/Bleualign/`
[12]`https://py-googletrans.readthedocs.io/`

| Id | Setting | sCOMET | dCOMET | LTCR-1 | LTCR-$1_f$ |
|----|---------|--------|--------|--------|--------|
| 1 | Sentence-level | 83.78 | 6.55 | 80.27 | 88.78 |
| 2 | 1 + Short-Term Memory | 84.50 | 6.68 | 77.89 | 87.41 |
| 3 | 2 + Long-Term Memory | 84.54 | 6.67 | 79.23 | 89.44 |
| 4 | 3 + Source Summary | 84.61 | 6.68 | 76.09 | 91.25 |
| 5 | 3 + Target Summary | 84.70 | 6.72 | 82.14 | 92.86 |
| 6 | 3 + Bilingual Summary | **84.72** | **6.74** | 82.49 | 93.60 |
| 7 | **6 + Record (DELTA)** | 84.70 | 6.72 | **86.44** | **95.25** |

Table 4: Ablation Study.

component and the effects of hyper-parameters, please refer to Appendix E. When provided with context sentences (Model 2), the model exhibits improved translation quality scores, but no significant enhancement in translation consistency is observed. The introduction of long-term memory contributes to more consistent translations (Model 3). Incorporating bilingual summaries (Model 6) led to an increase in COMET scores as well as consistency metrics, indicating that this component not only enhances translation quality but also reinforces translation consistency. When proper noun records are introduced (Model 7), a slight decrease in sCOMET and dCOMET scores is observed, likely due to the perturbation introduced by incorporating additional information. However, translation consistency improves significantly, with LTCR-1 increasing by 3.95 points and LTCR-$1_f$ increasing by 1.65 points compared to Model 6. Moreover, it is evident that the bilingual summary has a superior impact on both translation quality and consistency compared to using a summary on either the source side or the target side alone. Among Models 4, 5, and 6, Bilingual Summary achieves the highest scores across all four metrics.

**Consistency Distance** One significant challenge in document-level translation is maintaining long-term consistency. To evaluate whether our approach addresses this challenge, we divide the sentence-wise distance between each proper noun's translation and its first occurrence into several intervals. We then report the proportion of consistent translations in each interval in En $\Rightarrow$ Xx (upper) and Xx $\Rightarrow$ En (lower) in Figure 2. We observe that our approach almost outperforms the Sentence and Context methods in achieving proper noun translation consistency across all distance intervals. Notably, our approach excels when the distances exceed 50 sentences, yielding a larger proportion of consistent translations than the baseline methods. This demonstrates the effectiveness of our approach in enhancing long-context translation consistency.

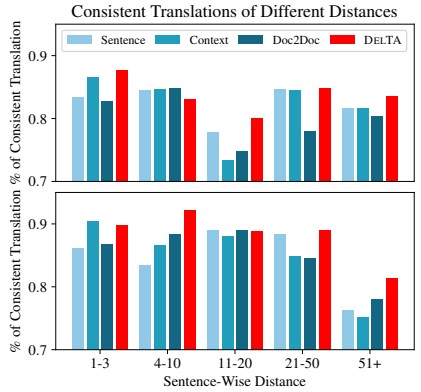

Figure 2: Proportions of consistent translations in different sentence-wise distances.

**Pronoun & Context-Dependent Translation** We follow Miculicich et al. (2018); Tan et al. (2019); Lyu et al. (2021) to evaluate the accuracy of pronoun translation (APT) of our system in En $\Rightarrow$ Zh using the reference-based metric proposed by Miculicich Werlen & Popescu-Belis (2017). We also evaluate our system on the first 1000 instances in the En $\Rightarrow$ De subset "mini.gender.opensubtitles" of a context-dependent translation benchmark called CTX-PRO Wicks & Post (2023). The results achieved by `GPT-3.5-Turbo-0125` are demonstrated in Table 5 and Table 6. DELTA improves the performance of pronoun

| Metric | Sentence | Context | Doc2Doc | DELTA |
|--------|----------|---------|---------|-------|
| APT | 59.96 | 60.84 | 56.11 | **61.07** |

Table 5: Evaluation results of pronoun translation accuracy (APT).

| Metric | Sentence | DELTA |
|--------|----------|-------|
| Generative Accuracy (%) | 29.7 | **51.0** |

Table 6: Evaluation results of context-dependent translation.

translation compared to the Sentence and Context baselines, and enhances translations where context information is explicitly needed. These results indicate that the multi-level memory in DELTA is beneficial to resolving coreference and discourse issues in the document.

**DELTA as a Summarize Writer**   To assess the effectiveness of our summary component, we conduct an experiment on QMSum (Zhong et al., 2021), a benchmark for query-based summarization, where systems are required to generate summaries of the meeting transcripts in response to a specified query. In our experiment, the query is incorporated into the prompts for both summary generation and the merging process. We compare our results with those of Lee et al. (2024), who paginate the document, generate summaries for each page, and then perform a lookup process on these summaries according to the query. The results are shown in Table 7. Our system's segment-by-segment summarization approach enables us to effectively locate relevant portions of the document and synthesize the information through the summary merging process. This demonstrates that the summary component of DELTA is also well-suited for general summarization tasks.

| System | ROUGE-L | Length |
|---|---|---|
| READAENT | 21.50 | 67.86 |
| DELTA | 23.60 | 82.28 |

Table 7: QMSum test results of ReadAgent and DELTA. "Length" denotes the word-wise length of the response.

**Memory Costs**   Our agent system consumes less memory than LLM-based Doc2Doc methods, such as those described by Wang et al. (2023b), where the whole document is translated within a single chat box, suffering from significant memory costs.   As shown in Figure 3, we compared the memory usage by utilizing `Qwen2-72B-Instruction` to translate a document in En $\Rightarrow$ Zh on a device with 2 NVIDIA A800 80GB GPUs. Our method demonstrates relatively slow memory growth as the number of processed sentences increases, primarily due to the increasing length of the summaries. In contrast, the memory usage of the Doc2Doc method increases significantly as the number of processed sentences grows, ultimately leading to memory exhaustion when the count reaches 490. This indicates that our approach is more memory-efficient and cost-effective for deployment on local devices.

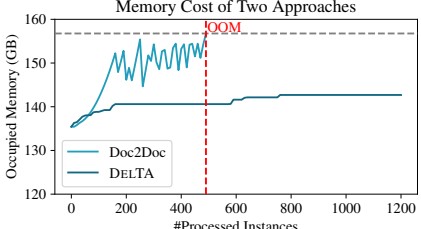

Figure 3: Memory cost of the Doc2Doc approach and our approach.

## 7   CONCLUSION

In this paper, we analyze two critical challenges for DocMT-LLMs: translation inconsistency and inaccuracy. We design an online document-level translation agent equipped with a multi-level memory component to tackle them. This memory structure retrieves and stores key information to assist the translation process, significantly enhancing translation consistency and quality. Notably, the effectiveness in maintaining proper noun translation consistency is particularly pronounced in novel translation, and our approach is still able to maintain consistency even when there is a large distance between the occurrences of a proper noun pair. The sentence-by-sentence online translation method avoids sentence omissions and reduces GPU memory consumption, in contrast to mainstream Doc2Doc approaches. Further analysis indicates that our framework can model document discourse structures to improve pronoun translation and context-dependent translation accuracy and the summarizer component in our agent is also capable of the query-based summarization task.

## LIMITATIONS

In this work, we present a framework for the DocMT agent, without prioritizing its inference efficiency. Given the complexity of DELTA's inference process, LLMs are frequently invoked during document translation, leading to prolonged runtime. To address this issue, we identify several potential directions for improvement. First, by explicitly marking sentence boundaries with special boundary tags, we can enforce sentence-level alignment within generated paragraphs, allowing LLMs to process multiple sentences concurrently and thereby reduce invocation frequency. Second, employing more precise alignment tools and scripts to extract proper nouns and their translations, rather than relying on LLMs, can further enhance the efficiency of DELTA. Other components, such as Long-Term Retriever, could be implemented using a dense retriever rather than employing LLMs to identify related sentences. Finally, in the summary component, reducing the summary generation to a single step by directly merging sentences within the window into an overall summary can also decrease runtime. We consider these optimizations to be future directions for our work.

ACKNOWLEDGMENTS

This work was supported in part by the National Natural Science Foundation of China (Grant No. 62206076), Guangdong Basic and Applied Basic Research Foundation (Grant No. 2024A1515011491), Shenzhen Science and Technology Program (Grant Nos. ZDSYS20230626091203008, KQTD2024072910215406, KJZD20231023094700001). Derek F. Wong was supported in part by the Science and Technology Development Fund of Macau SAR (Grant Nos. 0007/2024/AKP, FDCT/0070/2022/AMJ, FDCT/060/2022/AFJ), and the UM and UMDF (Grant Nos. MYRG-GRG2023-00006-FST-UMDF, MYRG-GRG2024-00165-FST-UMDF, EF2024-00185-FST, EF2023-00151-FST, EF2023-00090-FST). We would like to thank the anonymous reviewers and meta-reviewer for their insightful suggestions.

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

| | |
|---|---|
| **Proper Noun Translation Inconsistency** | |
| SRC | It's a story about this woman, Natalia Rybczynski. |
| HYP | 这是关于这个女人娜塔莉亚·雷布琴斯基的故事。 |
| SRC | Natalia Rybczynski: Yeah, I had someone call me "Dr. Dead Things." |
| HYP | 娜塔莉娅·丽琴斯基：是的，有人叫我"死物博士"。 |
| **Undertranslation** | |
| SRC | But here's the truth.//Here's the epiphany that I had that changed my thinking.//From 1970 until today, the percentage of the world's population living in starvation levels... |
| HYP | 但事实是，**// \*Missing translation\* //**自1970年至今，生活在饥饿水平、每天生活在一美元以下（当然要根据通货膨胀调整）的全球人口比例下降了80％。 |
| **Low Translation Quality** | |
| SRC | And we make decisions about where to live, who to marry and even who our friends are going to be, based on what we already believe. |
| REF | 我们做的各种决定，选择生活在何处，与谁结婚甚至和谁交朋友，都只基于我们已有的信念。 |
| HYP₁ | 我们根据自己已有的信念来做决定，包括选择居住的地方，结婚对象，甚至决定谁会成为我们的朋友。 |
| HYP₅₀ | 我们决定居住地、婚姻对象，甚至我们的朋友根据我们已经相信的事情。 |

Table 8: Demonstrations of proper noun translation inconsistency, undertranslation, and low translation quality issues encountered during document-level translation. In the first part, texts highlighted in red represent the same source proper noun with two different translations in the target document. In the second part, "//" indicates sentence boundaries in the document, illustrating that the translation of the second source sentence is absent from the target document. In the third part, "HYP$_n$" represents the hypothesis generated by the LLM using a translation window of size $n$.

## A  EXAMPLES OF TRANSLATION INCONSISTENCY AND INACCURACY

Examples of translation inconsistency, undertranslation, and low translation quality issues are presented in Table 8. In the first instance, the name "Natalia Rybczynski", which appears twice in the source document, is translated into two different forms: "娜塔莉亚·雷布琴斯基" and "娜塔莉亚·丽琴斯基". This variation leads to a notable inconsistency in the translation. In the second instance, the second sentence in the source document is omitted. Its corresponding translation is absent in the target document. This exemplifies an undertranslation issue in the document translation task. In the third instance, the translation produced by the LLM with a translation window of 50 sentences deviates significantly from the customary word order in Chinese compared to the sentence-level approach (using a window size of 1).

## B  DETAILED STATISTICS OF THE DATASETS IN OUR EXPERIMENTS

We conduct our experiments on two test sets: IWSLT2017 and Guofeng. The IWSLT2017 test set consists of parallel documents sourced from TED talks, covering 12 language pairs. Our experiments are conducted on eight of these pairs, En ⇔ Zh, De, Fr, and Ja. Each language pair contains 10 to 12 sentence-level aligned parallel documents, totaling approximately 1.5K sentences, with an average of around 120 sentences per document. Detailed statistics for each language pair are shown in the first block of Table 9.

The Guofeng Webnovel corpus is a high-quality and discourse-level corpus of web fiction, and we conduct our experiments on the Guofeng V1 TEST_2 set, which is designed in the Zh ⇒ En language pair. The second block of Table 9 illustrated the statistics of the test sets.

## C  PROMPT TEMPLATES FOR LLM-BASED COMPONENTS IN DELTA

This part details the prompts used for each module of DELTA. The prompt template for the Proper Noun Extractor is depicted in Figure 4. We use a prompt in the few-shot style to ensure accurate and formatted outputs. The prompt templates of the source and target summary writers are shown in

| Dataset | Language | $|S|$ | $|D|$ | $|S|/|D|$ |
|---------|----------|-----|-----|-----------|
| IWSLT2017 | Zh ⇔ En | 1459 | 12 | 122 |
| | De ⇔ En | 1138 | 10 | 114 |
| | Fr ⇔ En | 1455 | 12 | 121 |
| | Ja ⇔ En | 1452 | 12 | 121 |
| Guofeng V1 TEST_2 | Zh ⇒ En | 857 | 12 | 71 |

Table 9: Statistics of the test sets used in our experiments. "$|S|$" represents the number of sentences in each test set, "$|D|$" represents the number of documents, and $|S|/|D|$ represents the average number of sentences per document.

| Id | Setting | sCOMET | dCOMET | LTCR-1 | LTCR-1$_f$ |
|----|---------|--------|--------|--------|------------|
| 1 | Sentence-level | 83.78 | 6.55 | 80.27 | 88.78 |
| 2 | 1 + Short-Term Memory | 84.50 | 6.68 | 77.89 | 87.41 |
| 3 | 1 + Long-Term Memory | 84.48 | 6.69 | 78.77 | 88.01 |
| 4 | 1 + Record | 84.11 | 6.60 | 81.33 | 89.33 |
| 5 | 1 + Summary | 84.51 | 6.73 | 79.73 | 90.70 |
| 6 | 2 + Long-Term Memory | 84.54 | 6.67 | 79.23 | 89.44 |
| 7 | 2 + Record | 84.45 | 6.70 | 82.37 | 92.54 |
| 8 | 3 + Source Summary | 84.61 | 6.68 | 76.09 | 91.25 |
| 9 | 3 + Target Summary | 84.70 | 6.72 | 82.14 | 92.86 |
| 10 | 3 + Bilingual Summary | **84.72** | **6.74** | 82.49 | 93.60 |
| 11 | **10 + Record (DELTA)** | 84.70 | 6.72 | **86.44** | **95.25** |

Table 10: More detailed results of the ablation study.

Figure 5 and Figure 6, respectively. Note that the prompts for the summary components are written in their respective source or target language to avoid off-target issues. The prompt template for the Memory Retriever is shown in Figure 7. The prompt template for the Document Translator is illustrated in Figure 8. All the retrieved information from the multi-level memory components is formatted into this prompt to assist the translation process.

## D    DETAILED RESULTS OF THE MAIN EXPERIMENT

The scores for the En ⇒ Zh, De, Fr, Ja translation directions are presented in Table 13, while the scores for the Zh, De, Fr, Ja ⇒ En are shown in Table 14.

DELTA achieves improvements in both translation consistency, as indicated by the LTCR-1 and LTCR-1$_f$ metrics, and translation quality, as indicated by the sCOMET and dCOMET metrics, across most translation directions compared to several baselines. The Qwen models show significant enhancements in sCOMET and dCOMET scores, demonstrating that our approach provides strong reinforcement for the document translation quality of these open-source models.

The quality improvements are most pronounced in the Ja ⇔ En directions across all backbone models, likely due to their modest baseline capabilities for these language pairs, which our approach effectively enhances. Translation consistency in the Zh ⇔ En directions benefits most from our approach, as the distinct character sets of Chinese and English pose challenges in maintaining proper noun consistency, highlighting the effectiveness of our method.

When applying GPT-3.5-Turbo as the backbone model, DELTA outperforms translation-specialized baselines, such as NLLB-3.3B and GoogleTrans, across most languages, demonstrating the promising capabilities of LLM-based autonomous agents in document translation.

## E    DETAILED RESULTS OF THE ABLATION STUDY

| Short. Window Size | sCOMET | dCOMET | LTCR-1 | LTCR-1$_f$ |
|---|---|---|---|---|
| 1 | 84.51 | 6.71 | 87.42 | 95.36 |
| 3 | 84.70 | 6.72 | 86.44 | 95.25 |
| 5 | 84.65 | 6.74 | 87.42 | 95.03 |
| Long. Window Size | sCOMET | dCOMET | LTCR-1 | LTCR-1$_f$ |
| 10 | 84.61 | 6.74 | 86.67 | 95.67 |
| 20 | 84.70 | 6.72 | 86.44 | 95.25 |
| 30 | 84.71 | 6.73 | 84.25 | 94.86 |
| Growing | 84.68 | 6.71 | 85.57 | 93.96 |

Table 11: Effect of the Short-Term and Long-Term window size.

| System | sCOMET | dCOMET | LTCR-1 | LTCR-1$_f$ |
|---|---|---|---|---|
| Sentence | 87.33 | 7.04 | 78.65 | 96.07 |
| Context | 87.87 | 7.19 | 78.09 | 96.07 |
| DELTA | 87.96 | 7.20 | 82.68 | 96.65 |

Table 12: Evaluation results on the Lt $\Rightarrow$ En low-resource test set.

**Effect of Each Memory Component**    The ablation results for each individual memory component are presented in Table 10. The backbone model used in our experiments is GPT-3.5-Turbo, evaluated on the IWSLT2017 En $\Rightarrow$ Zh test set. We can observe that 1) Short-Term Memory enhances translation quality but negatively impacts consistency due to disruptions caused by the short-span context it introduces (Model 2). 2) Similarly, Long-Term Memory improves translation quality but also leads to a decline in consistency (Model 3). However, these two modules complement each other, as their integration leverages information from different spans, resulting in further quality enhancements while mitigating consistency issues (Model 6). 3) Proper Noun Records contribute to both translation quality and consistency, as improved accuracy in proper noun translation directly enhances overall quality (Model 4). 4) Bilingual Summary also positively impacts translation quality by introducing the document's main idea, which guides the generation of target sentences more effectively (Model 5).

Each module independently contributes to performance improvements. However, none surpass the combination of all modules working together. We also conducted an additional comparative test between DelTA and the context method augmented with Proper Noun Records. The results, presented as Model 7 in Table 10, reveal a large performance gap favoring DelTA. We attribute this to the DelTA architecture's ability to incorporate information across varying granularities and scales, enabling a synergistic enhancement of both translation quality and consistency within our framework.

**Effect of Hyper-Parameters**    We perform experiments to evaluate the impact of different window sizes for Short-Term and Long-Term Memory (i.e. $k$ and $l$ introduced in §4). The results are presented in Table 11. Increasing the window size of Short-Term Memory results in a slight improvement in consistency but incurs a substantial additional computational cost, as two sentence pairs extend each sentence's translation prompt. Given the trade-off between computational cost, translation quality, and consistency, we opted to employ a Short-Term window size of 3. Further increasing the Long-Term Memory window does not yield significant benefits. To address the idea that the model might require longer context windows in the later stages of document translation, we also experimented with a dynamic window setting, where the window size increases by one sentence pair for every eight sentences translated. However, this approach does not lead to performance improvements. We attribute this to the Bilingual Summary component, which effectively captures key information from more distant contexts. The iterative process of summary generation and merging inherently functions as a form of context window growth, rendering additional adjustments to the Long-Term Memory window unnecessary.

## F   PERFORMANCE ON LOW-RESOUCE LANGUAGES

We evaluate how well DELTA scales to low-resource languages. Considering that the spaCy NLP tool used for evaluation does not support many low-resource languages, we select Lt ⇒ En as the language pair to test. We randomly sample a document (with 556 sentences) from Europarl v9 training set[13] as our evaluation set, and employ GPT-3.5-Turbo as the backbone model. The results are shown in Table 12, indicating that DELTA also performs well on the low-resource language pair.

---

**Prompt for Proper Noun Extractor**

You are an English-Chinese bilingual expert. Given an English source sentence with its Chinese translation, you need to annotate all the proper nouns in the English source sentence and their corresponding translations in the Chinese translation sentence. Here are some examples for you:

Example 1:
<English source> NASA's Kepler mission has discovered thousands of potential planets around other stars, indicating that Earth is but one of billions of planets in our galaxy.
<Chinese translation> 美国国家航空航天局的开普勒任务已经发现了围绕着其他恒星的数千颗潜在的行星，这也表明了地球只是银河系中数十亿行星中的一颗。
<Proper nouns> "NASA" - "美国国家航空航天局", "Kepler" - "开普勒", "Earth" - "地球"

Example 2:
<English source> I had just driven home, it was around midnight in the dead of Montreal winter, I had been visiting my friend, Jeff, across town, and the thermometer on the front porch read minus 40 degrees – and don't bother asking if that's Celsius or Fahrenheit, minus 40 is where the two scales meet – it was very cold.
<Chinese translation> 我开车回到家，在Montreal的寒冬，大约午夜时分，我开车从城镇一边到另一边，去看望我的朋友杰夫，门廊上的温度计显示零下40度——不需要知道是摄氏度还是华氏度，到了零下40度，两个温度显示都一样——天气非常冷。
<Proper nouns> "Montreal" - "N/A", "Jeff" - "杰夫", "Celsius" - "摄氏度", "Fahrenheit" - "华氏度"

Example 3:
<English source> To make the case to the National Health Service that more resources were needed for autistic children and their families, Lorna and her colleague Judith Gould decided to do something that should have been done 30 years earlier.
<Chinese translation> 为了向国家医疗保健系统证明，自闭症儿童和他们的家庭需要更多的资源，Lorna和她的同事朱迪思·古尔德决定去做一些三十年前就应该被完成的事情。
<Proper nouns> "National Health Service" - "国家医疗保健系统", "Lorna" - "N/A", "Judith Gould" - "朱迪思·古尔德"

If there isn't any proper noun in the sentence, just answer with "N/A". Now annotate all the proper nouns in the following sentence pair:
<English source> {SOURCE_SENTENCE}
<Chinese translation> {TARGET_SENTENCE}
<Proper nouns>

---

Figure 4: Prompt template for the Proper Noun Extractor. We provide several few-shot exemplars preceding the current input. This template is designed for the En ⇒ Zh translation direction. For other translation directions, adjust the corresponding content to match the specific languages.

---

[13] https://www.statmt.org/europarl/v9/training/

---

**Prompt for Source Summary Writer (Segment Summary Generation)**

Below is a paragraph. Please provide a summary of this paragraph, including the main contents of these sentences, the overall domain, style and tone of it, while preserving key information as much as possible.
Paragraph: {SOURCE_SEGMENT}
Summary:

---

**Prompt Template for Source Summary Writer (Summary Merging)**

Below are the summaries of two adjacent paragraphs. Please merge them into a single summary, retaining as much key information as possible and ensuring that information about the domain, style, and tone are preserved.
Summary 1: {SUMMARY_1}
Summary 2: {SUMMARY_2}
Merged summary:

---

Figure 5: Prompt template for Source Summary Writer.

---

**Prompt Template for Target Summary Writer (Segment Summary Generation)**

Below is a paragraph. Please provide a summary of this paragraph, while preserving key information as much as possible.
Paragraph: {SOURCE_SEGMENT}
Summary:

---

**Prompt Template for Target Summary Writer (Summary Merging)**

Below are the summaries of two adjacent paragraphs. Please merge them into a single summary, retaining as much key information as possible.
Summary 1: {SUMMARY_1}
Summary 2: {SUMMARY_2}
Merged summary:

---

Figure 6: Prompt template for Target Summary Writer. We write the prompt in the target language to better align with the agent profile of the monolingual summary writer and reduce the off-target issues. For demonstration purposes, the prompts provided here are written in English.

---

**Prompt Template for Long-Term Memory Retriever**

You are a linguistic expert. Given a list of sentences and a query, your task is to find the {TOP_NUM} sentences in the list that are most relevant to the request.

Sentence list:
{SRC1}
{SRC2}
. . .

Query:
{QUERY}

Note that you should only respond with a list containing the numbers of these {TOP_NUM} sentences. For example, if you choose sentences 15, 16, and 19 as your answer, your response should be "[15, 16, 19]".

---

Figure 7: Prompt template for Long-Term Memory Retriever.

---

**Prompt Template for Document Translator**

You are an {SRC_LANG}-{TGT_LANG} bilingual expert, translating a very long {SRC_LANG} document. Given the summary of the preceding text in both {SRC_LANG} and {TGT_LANG}, the historical translation of some proper nouns, source and translation texts preceding the current sentence, as well as some relevant translation instances from the preceding text, translate the current {SRC_LANG} source sentence into {TGT_LANG}. Please ensure that the translations of proper nouns in the source sentence are consistent with their historical translation, and the translation style remains consistent as well.

Summaries:
<{SRC_LANG} summary> {SRC_SUMMARY}
<{TGT_LANG} summary> {TGT_SUMMARY}

Historical translations of proper nouns:
{HISTORY}

Preceding texts:
<{SRC_LANG} text> {SRC_CONTEXT}
<{TGT_LANG} text> {TGT_CONTEXT}

Relevant instances:
{RELEVANT_INSTANCES}

Now translate the following {SRC_LANG} source sentence to {TGT_LANG}.
<{SRC_LANG} source> {SOURCE}
<{TGT_LANG} translation>

---

Figure 8: Prompt template for Document Translator. Involved proper nouns and their corresponding translations are formatted into the "HISTORY" field. Source and target sentences from Short-Term Memory are concatenated and formatted into the "SRC_CONTEXT" and "TGT_CONTEXT" fields, respectively. Retrieved instances from Long-Term Memory are formatted into the "RELE-VANT_INSTANCES" field as source-target pairs.

| System | sCOMET | dCOMET | LTCR-1 | LTCR-$1_f$ | sCOMET | dCOMET | LTCR-1 | LTCR-$1_f$ |
|---|---|---|---|---|---|---|---|---|
| | | En $\Rightarrow$ Zh | | | | En $\Rightarrow$ De | | |
| NLLB | 76.81 | 6.20 | 71.68 | 84.96 | 84.15 | 6.64 | 91.76 | 99.61 |
| GOOGLE | 78.46 | 5.76 | 89.45 | 92.36 | 80.23 | 5.78 | 93.55 | 99.19 |
| GPT-3.5-Turbo | | | | | | | | |
| Sentence | 83.78 | 6.55 | 80.27 | 88.78 | 84.97 | 6.71 | 92.06 | 98.81 |
| Context | 84.50 | 6.68 | 77.89 | 87.41 | 85.12 | 6.74 | **93.70** | 99.21 |
| Doc2Doc | – | 6.29 | 82.04 | 94.29 | – | **6.81** | 88.89 | 97.94 |
| Ours | **84.70** | **6.72** | **86.44** | **95.25** | **85.37** | 6.78 | 93.46 | **99.23** |
| GPT-4o-mini | | | | | | | | |
| Sentence | 82.13 | 6.43 | 78.04 | 91.89 | 81.41 | 6.39 | 90.70 | 98.84 |
| Context | 84.36 | 6.68 | 78.95 | **93.42** | 84.83 | 6.70 | **92.66** | 99.61 |
| Doc2Doc | – | 6.60 | 82.33 | 88.35 | – | **6.90** | 91.05 | 99.22 |
| Ours | **84.94** | **6.81** | **85.52** | 91.72 | **85.47** | 6.79 | 92.19 | **100.0** |
| Qwen-7B-Instruct | | | | | | | | |
| Sentence | 83.05 | 6.51 | 77.78 | 83.84 | 76.24 | 5.54 | 82.11 | 90.65 |
| Context | 83.51 | 6.67 | 77.29 | 87.12 | 76.67 | **5.56** | 85.66 | 92.45 |
| Doc2Doc | – | 6.16 | **81.85** | **91.11** | – | 5.25 | **88.60** | **97.93** |
| Ours | **83.98** | **6.70** | 80.13 | 90.55 | **76.84** | **5.56** | 87.40 | 96.75 |
| Qwen-72B-Instruct | | | | | | | | |
| Sentence | 77.76 | 5.88 | 73.13 | 83.96 | 78.98 | 6.13 | **92.18** | 98.35 |
| Context | 81.69 | 6.24 | 78.01 | 86.17 | 80.96 | 6.43 | 89.64 | 96.81 |
| Doc2Doc | – | 6.22 | 74.59 | 78.38 | – | 6.66 | 88.28 | **98.44** |
| Ours | **84.76** | **6.70** | **81.56** | **89.72** | **84.29** | **6.70** | 90.48 | 98.41 |
| | | En $\Rightarrow$ Fr | | | | En $\Rightarrow$ Ja | | |
| NLLB | 85.35 | 6.23 | 87.84 | 89.86 | 82.12 | 6.37 | 46.94 | 53.06 |
| GOOGLE | 82.21 | 5.53 | 88.61 | 91.46 | 80.74 | 6.23 | 53.92 | 55.88 |
| GPT-3.5-Turbo | | | | | | | | |
| Sentence | 85.84 | 6.18 | 83.55 | 88.49 | 84.61 | 6.89 | 52.34 | 55.14 |
| Context | **86.49** | 6.27 | 83.06 | 89.25 | 85.50 | 7.09 | 54.72 | 56.60 |
| Doc2Doc | – | 6.28 | **92.28** | **94.63** | – | 7.10 | 53.26 | 58.70 |
| Ours | 86.48 | **6.30** | 88.96 | 94.16 | **85.76** | **7.13** | **62.96** | **66.67** |
| GPT-4o-mini | | | | | | | | |
| Sentence | 80.79 | 5.82 | 88.89 | 90.91 | 81.72 | 6.74 | 56.73 | 58.65 |
| Context | 85.10 | 6.14 | 90.52 | 92.81 | 84.84 | 7.09 | 57.89 | **62.11** |
| Doc2Doc | – | 6.24 | **91.00** | **94.00** | – | 7.25 | 57.78 | 60.00 |
| Ours | **86.38** | **6.28** | 90.94 | 93.85 | **86.61** | **7.32** | **58.54** | 59.76 |
| Qwen-7B-Instruct | | | | | | | | |
| Sentence | 80.61 | 5.47 | 82.53 | 84.93 | 80.21 | 6.30 | 53.21 | 58.72 |
| Context | **81.31** | **5.54** | **89.00** | 90.72 | 81.85 | 6.53 | **66.41** | 71.09 |
| Doc2Doc | – | 5.29 | 87.88 | **93.56** | – | **6.63** | 50.96 | 55.77 |
| Ours | 81.02 | 5.46 | 88.66 | 91.41 | **82.23** | 6.57 | 64.18 | **72.39** |
| Qwen-72B-Instruct | | | | | | | | |
| Sentence | 81.02 | 5.75 | 90.00 | 92.33 | 76.34 | 6.11 | **62.86** | 65.71 |
| Context | 84.03 | 6.06 | 87.22 | 89.46 | 76.48 | 6.14 | 61.68 | 69.16 |
| Doc2Doc | – | 6.25 | 89.25 | 91.86 | – | 6.66 | 42.20 | 45.87 |
| Ours | **85.76** | **6.28** | **92.08** | **94.39** | **85.13** | **6.94** | 62.50 | **70.83** |

Table 13: Detailed results of our experiments in En $\Rightarrow$ Xx directions.

| System | sCOMET | dCOMET | LTCR-1 | LTCR-1$_f$ | sCOMET | dCOMET | LTCR-1 | LTCR-1$_f$ |
|---|---|---|---|---|---|---|---|---|
| | | | Zh $\Rightarrow$ En | | | | De $\Rightarrow$ En | |
| NLLB | 82.14 | 7.01 | 75.31 | 88.27 | 85.63 | 7.23 | 95.98 | 98.85 |
| Google | 78.06 | 5.68 | 72.89 | 86.14 | 82.31 | 6.47 | 96.53 | 98.27 |
| GPT-3.5-Turbo | | | | | | | | |
| Sentence | 83.34 | 7.17 | 73.99 | 86.71 | 85.92 | 7.26 | **98.88** | **100.0** |
| Context | **83.88** | 7.29 | 76.92 | 90.53 | 86.10 | **7.30** | 98.30 | **100.0** |
| Doc2Doc | – | 7.08 | 76.77 | 88.39 | – | 7.16 | 98.24 | 98.82 |
| Ours | 83.88 | **7.30** | **80.00** | **93.53** | **86.14** | **7.30** | 98.33 | **100.0** |
| GPT-4o-mini | | | | | | | | |
| Sentence | 83.55 | 7.24 | 71.93 | 84.80 | 85.11 | 7.17 | 98.20 | **100.0** |
| Context | 83.96 | 7.35 | 78.24 | 91.18 | 86.12 | 7.27 | **98.88** | **100.0** |
| Doc2Doc | – | 7.15 | **79.62** | 92.36 | – | 7.19 | 95.24 | 97.62 |
| Ours | **84.10** | **7.47** | 79.41 | **94.71** | **86.61** | **7.31** | 98.32 | **100.0** |
| Qwen-7B-Instruct | | | | | | | | |
| Sentence | 79.44 | 6.76 | 71.17 | 87.73 | 77.20 | 6.75 | 93.25 | 95.09 |
| Context | 82.62 | 7.04 | 69.70 | 83.64 | 84.52 | **7.08** | **98.80** | 99.40 |
| Doc2Doc | – | 6.53 | **82.79** | **92.62** | – | 6.88 | 98.67 | 99.33 |
| Ours | **82.83** | **7.09** | 76.47 | 92.35 | **84.61** | 7.05 | 98.25 | **100.0** |
| Qwen-72B-Instruct | | | | | | | | |
| Sentence | 80.35 | 6.95 | 73.49 | 84.34 | 81.17 | 6.94 | 97.19 | **100.0** |
| Context | 80.11 | 6.93 | 74.25 | 86.23 | 84.81 | 7.24 | **98.31** | 99.44 |
| Doc2Doc | – | 6.94 | 68.21 | 80.13 | – | 7.08 | 97.59 | 98.19 |
| Ours | **84.51** | **7.40** | **83.93** | **94.05** | **86.17** | **7.34** | 98.29 | **100.0** |
| | | | Fr $\Rightarrow$ En | | | | Ja $\Rightarrow$ En | |
| NLLB | 87.59 | 6.79 | 93.56 | 97.42 | 81.02 | 6.90 | 51.27 | 78.48 |
| Google | 84.64 | 6.19 | 95.63 | 96.83 | 75.67 | 5.50 | 60.67 | 82.00 |
| GPT-3.5-Turbo | | | | | | | | |
| Sentence | 87.60 | 6.78 | 94.94 | 97.89 | 81.00 | 6.98 | 60.12 | 82.82 |
| Context | **88.03** | 6.84 | 94.96 | 97.90 | **81.85** | **7.17** | 69.94 | 92.64 |
| Doc2Doc | – | 6.78 | 94.78 | 97.39 | – | 6.80 | 70.90 | 87.31 |
| Ours | 88.02 | **6.86** | **96.17** | **98.30** | 81.76 | 7.13 | **71.60** | **93.21** |
| GPT-4o-mini | | | | | | | | |
| Sentence | 87.32 | 6.77 | 94.42 | 97.85 | 80.04 | 6.76 | 61.11 | 82.72 |
| Context | 87.72 | 6.81 | 94.85 | 97.42 | 82.00 | 7.17 | 65.62 | 88.75 |
| Doc2Doc | – | 6.83 | 94.42 | 98.28 | – | 6.86 | 64.71 | 85.29 |
| Ours | **88.13** | **6.90** | **96.58** | **98.72** | **82.20** | **7.29** | **66.67** | **90.12** |
| Qwen-7B-Instruct | | | | | | | | |
| Sentence | 81.21 | 6.27 | 87.90 | 95.56 | 70.56 | 6.15 | 53.25 | 73.38 |
| Context | **86.55** | **6.63** | **94.19** | 97.51 | 78.68 | 6.59 | 63.23 | **89.68** |
| Doc2Doc | – | 6.57 | 87.00 | **97.76** | – | 6.39 | **71.67** | 85.00 |
| Ours | 86.40 | 6.56 | 91.20 | 92.80 | **79.60** | **6.66** | 62.26 | 88.05 |
| Qwen-72B-Instruct | | | | | | | | |
| Sentence | 82.67 | 6.38 | 95.44 | 98.34 | 77.94 | 6.63 | 62.89 | 85.53 |
| Context | 87.02 | 6.76 | 93.25 | 97.89 | 81.13 | 7.02 | 65.64 | 85.28 |
| Doc2Doc | – | 6.73 | 94.67 | 97.78 | – | 6.74 | **71.53** | 86.86 |
| Ours | **88.01** | **6.87** | **95.78** | **98.73** | **82.06** | **7.24** | 68.12 | **93.12** |

Table 14: Detailed results of our experiments in Xx $\Rightarrow$ En directions.

