# OpenReview forum: "DelTA: An Online Document-Level Translation Agent Based on Multi-Level Memory"
_ICLR.cc/2025/Conference — ICLR 2025 Poster_

### Official Review · Reviewer_gPC6 · 2024-10-28

**Soundness:** 3
**Presentation:** 3
**Contribution:** 3
**Rating:** 6
**Confidence:** 4

**Summary:**

The paper introduces DELTA, a Document-level Translation Agent that uses a multi-level memory architecture to improve translation consistency and quality in document-level machine translation. The agent employs four memory components (Proper Noun Records, Bilingual Summary, Long-Term Memory, and Short-Term Memory) that store and retrieve information across different granularities. The approach shows impressive improvements in translation consistency and quality across multiple language pairs and different LLM backbones. The work is evaluated extensively using both established metrics and novel consistency measures.

**Strengths:**

**Originality**:
- The multi-level memory architecture represents a novel approach to document translation. While memory architectures have been used in LLM agents before, the specific design with four complementary components (Proper Noun Records, Bilingual Summary, Long/Short-Term Memory) targeting different aspects of document translation is innovative.
- The approach of using an LLM-based agent for document translation, while maintaining sentence-level alignment, is creative and practical.
- The introduction of LTCR-1 and LTCR-1f metrics for evaluating translation consistency is a useful contribution to the field.

**Quality**:
- The experimental methodology is comprehensive, evaluating across 8 language pairs and multiple LLM backbones (both open and closed source).
- The ablation studies effectively demonstrate the contribution of each memory component.
- The memory efficiency analysis is well-done, with clear comparisons showing the advantages over Doc2Doc approaches.
- The evaluation considers both translation quality (sCOMET, dCOMET) and consistency (LTCR metrics).

**Clarity**:
- The paper is logically structured, moving from motivation to design to evaluation.
- The multi-level memory architecture is well-explained with clear diagrams and algorithms.
- The experimental results are presented systematically with detailed tables and analyses.
- The limitations and future work are honestly discussed.

**Significance**:
- The work addresses a real and important challenge in document translation - maintaining consistency while handling long documents.
- The approach is practical and implementable, with clear memory efficiency advantages.
- The method shows strong improvements across different languages and models, suggesting broad applicability.
- The memory architecture could potentially be adapted for other document-level NLP tasks, as demonstrated by the query-based summarization results.

**Weaknesses:**

1. Efficiency analysis needs more detail:
   - While memory efficiency is demonstrated, the paper claims cost-effectiveness without supporting evidence
   - No comparison of total wall-time or FLOPS between approaches
   - Limited discussion of computational overhead from multiple LLM calls

2. Missing comparisons and baselines:
   - No evaluation against long-context LLMs (e.g., Claude, Gemini) which could be particularly relevant for document translation
   - DeepL, which explicitly supports document-level translation, would be a valuable baseline, and a more relevant choice than Google.
   - Limited discussion of Doc2Doc approach limitations in related work

3. Incomplete motivation and analysis:
   - Does not address context-dependent translation as a key challenge (see https://aclanthology.org/2023.wmt-1.42/)
   - Claims of "significant" improvements lack statistical significance testing
   - Claims about undertranslation issues and translation quality decline (L177-178) should be supported with citations to relevant work (e.g.,  https://arxiv.org/abs/2401.06468, https://aclanthology.org/2024.acl-long.336/)

4. Technical details and reproducibility:
   - No mention of code release plans

**Questions:**

1. How does the approach perform compared to long-context LLMs? Given recent advances in context window sizes, this seems particularly relevant.

2. Will the code be released? This would be valuable for reproducibility and adoption.

3. Can you provide a more detailed efficiency analysis, including:
   - Total wall-time comparison between DELTA and Doc2Doc approaches
   - FLOPS comparison for open-weight models
   - Analysis of the number of LLM calls required

4. How does DELTA handle context-dependent translations where meaning depends on broader document context? While pronoun translation is briefly discussed, a more comprehensive analysis (see https://aclanthology.org/2023.wmt-1.42/) would be valuable.

5. Have you considered evaluating against DeepL, given its explicit support for document-level translation?

6. Could you provide statistical significance tests for the reported improvements?

---

> ### Author Response · Authors · 2024-11-21
> **Response to Reviewer gPC6**
>
> > Q1. Claims about undertranslation issues and translation quality decline (L177-178) should be supported with citations to relevant work.
>
> Thanks for your reminder. We have added two relative citations here in our paper.
>
> > Q2. Comparison with long-context LLMs and commercial document translation tools.
>
> We compare DelTA with Gemini-1.5-Flash, Claude-3.5-Haiku, and the translation tool of DeepL on IWSLT2017. The results are shown in the following tables:
>
> | IWSLT2017           | COMET | LTCR-1 | Runtime (s/sent) |
> | ------------------- | ----- | ------ | ---------------- |
> | Gemini              | 85.37 | 84.77  | 5.29             |
> | Claude              | 84.85 | 83.06  | 7.56             |
> | DeepL               | 85.34 | 77.22  | -                |
> | DelTA-GPT-3.5-Turbo | 84.70 | 86.44  | 3.41             |
> | DelTA-GPT-4o-mini   | 84.94 | 85.52  | 3.68             |
>
> We observe that DelTA outperforms Gemini, Claude, and DeepL in the consistency metric, highlighting its strength in maintaining translation consistency. However, DelTA lags slightly behind Gemini and DeepL in the quality metric, which we attribute to inherent limitations in bilingual capability. It is worth noting that the long-context capabilities of these models come with significantly higher GPU memory consumption and increased inference latency, which lead to higher deployment costs. Our work aims to enhance the long-document translation performance of models with smaller context windows and lower deployment costs, such as the open-sourced models.
>
> > Q3. Will the code be released?
>
> The codes and data are released in the ANONYMOUS repository: https://anonymous.4open.science/r/DelTA_Agent-7716.
> We have added this link in our abstract. Thanks for your feedback!
>
> > Q4. More detailed efficiency analysis.
>
> We apologize for any confusion caused. When we state in the paper that our system is memory-efficient, we mean that DelTA consumes significantly less overall GPU memory compared to the Doc2Doc method. The Doc2Doc approach utilizes previous texts and their translations as chat histories to leverage the LLM’s context capabilities. However, as the document length increases, this method leads to a rapid escalation in memory usage. In contrast, DelTA’s memory consumption grows at a much slower rate, primarily due to the summary extension mechanism. This makes DelTA a more practical and scalable framework compared to the Doc2Doc approach.
>
> > Q5. How does DelTA handle context-dependent translations?
>
> We conduct a comparison between the Sentence method and DelTA on the mentioned context-dependent translation test set called CTXPRO. Due to time constraints, we only take the first 1000 instances of the “mini.gender.opensubtitles” dataset as the test set and test our systems in the English-German direction. The results are shown in the following table:
>
> | System   | Generative Accuracy (%) |
> | -------- | ----------------------- |
> | Sentence | 29.7                    |
> | DelTA    | 51.0                    |
>
> We can observe that DelTA achieves higher generative accuracy compared to the sentence-level method, indicating the strong performance of our framework on the context-dependent translation task.
>
> We have added this experiment to Section 6 of our paper. Thanks for your suggestion!
>
> > Q6. Statistical significance tests.
>
> The p-values of t-tests between DelTA and Sentence/Context methods are demonstrated in the following table:
>
> |                   | En-Xx |        | Xx-En |        |
> | ----------------- | ----- | ------ | ----- | ------ |
> | System            | COMET | LTCR-1 | COMET | LTCR-1 |
> | DelTA vs Sentence | <0.05 | 0.3472 | <0.05 | <0.05  |
> | DelTA vs Context  | <0.05 | 0.5867 | <0.05 | 0.2855 |
>
> Note that our framework achieves more consistency improvements on language pairs that do not share common alphabets, because proper nouns just need to be copied into the target sentences during translation for other language pairs. The p-values of LTCR-1 in En <=> Zh directions are less than 0.05.
>
> We have added these results in Section 5.2 of our paper. Thanks for your feedback!

---

> > ### Comment · Reviewer_gPC6 · 2024-11-21
> >
> > Thanks for your response, that clears up most of the questions and I have raised my soundness score.
> >
> > I still think  a more detailed efficiency analysis would be a very beneficial addition. I do understand that DelTA consumes significantly less overall GPU memory compared to the Doc2Doc method, but it'd be nice to see some quantitative comparisons of that, as well as wall-time or FLOPS.

---

> ### Author Response · Authors · 2024-11-22
>
> Thanks for adjusting your scores.
>
> We compare the wall time taken by DelTA with that of the baseline approach that retains the translation of each sentence as a historical chat context. The evaluation employs Qwen2-7B-Instruct and Qwen2-72B-Instruct as the backbone models. A document consisting of 100 sentences, sampled from the IWSLT2017 dataset, is translated from English to Chinese to measure both the total wall time and the average wall time per sentence. We use 1x NVIDIA A800 80G GPU for Qwen2-7B-Instruct, and 2x NVIDIA A800 80G GPUs for Qwen2-72B-Instruct. The results are shown in the following table:
>
> |                    | Total (s) | Average (s/sent) |
> | ------------------ | --------- | ---------------- |
> | Qwen2-7B-Instruct  | | |
> | Baseline           | 47 | 0.47 |
> | DelTA              | 204 | 2.04 |
> | Qwen2-72B-Instruct | | |
> | Baseline           | 2183 | 21.83 |
> | DelTA              | 1213 | 12.13 |
>
> As observed, when the 7B model is used as the backbone, where GPU memory usage is relatively low, DelTA incurs a longer inference time per sentence compared to the baseline method. In contrast, when the 72B model is employed, the wall time of the baseline method exceeds that of DelTA, mainly due to the exhaustion of memory and the over-length of context. These results further demonstrate that our framework offers greater ease of deployment compared to the baseline, particularly when utilizing large-scale models.
>
> Furthermore, as part of our future work, we plan to explore case-by-case adjustments to the inference trajectory, dynamically determining whether specific components should be invoked based on the characteristics of individual sentences. Additionally, we aim to investigate chunk-by-chunk document translation to enhance the parallelism of the inference process. We believe these strategies will further improve the efficiency of our framework.

---

> > ### Comment · Reviewer_gPC6 · 2024-11-25
> >
> > Thanks for the additional numbers, the trade-offs are more clear now.

---

### Official Review · Reviewer_h7o9 · 2024-11-01

**Soundness:** 3
**Presentation:** 3
**Contribution:** 2
**Rating:** 6
**Confidence:** 3

**Summary:**

This paper proposes a LLM-based, document-level translation agent (called Delta). This agent performs document-level translation on a sentence-by-sentence basis, and consists of auxiliary "multi-level memory" components, which are updated after translation of each sentence. These memory components include Proper Noun Records, Bilingual Summary, and Long-Term/Short-Term Memory. The information stored in all of these components is retrieved by prompting LLMs, where the prompt for each component is targeted towards its desired function (e.g., the Bilingual Summary component asks the LLM to generate a summary given a chunk of sentences). During translation of each sentence within a document, after updating all of these memory components, the Delta system aggregates their stored information into a single prompt. This process is continued iteratively until the entire document is translated. The paper shows that the Delta system outperforms other document-level MT systems on the IWSLT2017 and Guofeng datasets, and also presents ablations to understand the contribution of each memory component to the overall performance of Delta.

**Strengths:**

1. Document-level translation is currently one of the most important and underexplored research areas in the field of machine translation (MT). This paper clearly presents the motivation for their work, and summarizes the importance of document-level MT.
2. The Delta system presented in the paper is easy to implement, since it just requires prompting of off-the-shelf LLMs. The motivation for each of the modules in the Delta system is clear.
3. The experimental setup is well-written and easy to follow. The results are comprehensive (covering multiple LLMs and language pairs), and comparisons against all essential document-level MT baselines are presented.

**Weaknesses:**

1. The paper mentions that inconsistency is one of the major challenges during document-level translation. However, the only type of inconsistency they measure and report is Proper Noun Inconsistency, which is a surface-level type of inconsistency (can the model consistently copy a fixed string?) that doesn't require the model to reason across long contexts. The results on Proper Noun Inconsistency are used to make strong claims about their Delta system's overall improvement in consistency relative to baselines, but these claims are not adequately supported with evidence.
2. The paper focuses on the comparison to "Doc2Doc" (comparing memory costs, etc), but from the reported results, "Context" is consistently better than "Doc2Doc". There is no significance testing for the results in the paper (e.g., Tables 2 and 3), and in terms of translation quality (as measured by sCOMET and dCOMET), the performance of the "Context" and "Delta" models looks quite close, with "Context" outperforming "Delta" according to dCOMET for the Qwen2-7B-Instruct model in Table 2. In Table 3, the quality-based performance of "Context" and "Delta" is even closer. (And it is not surprising that Delta outperforms in proper noun consistency, since the proper noun extractor module in Delta provides a list of all proper nouns seen. If the Context model were enhanced with this module, it seems likely that its performance on the proper noun consistency metric would also correspondingly improve.)
3. The Delta system is comprised of 5 auxiliary modules. The ablation to isolate the effects of each individual module in Figure 4 shows very small differences in quality (as measured by sCOMET and dCOMET) across these models (again, with no significance testing). The recipe (combination of modules) proposed by the authors works reasonably well based on their results, but the paper provides limited insight into the relative contribution of each of these modules to improving performance. For instance, does short-term or long-term memory help more? Is the Memory Retriever step for long-term memory necessary, and how much does this step improve performance over just using the (l sentences in) long-term memory?
See additional questions below.

**Questions:**

1. For the summary writers, why isn't the summary for chunk m conditioned on the summary for chunk m-1 (which would allow the summary to take longer context into account)?
2. Did you investigate the effect of adjusting the numbers (k, l) of sentences to use as short- and long-term memory throughout the translation process? It seems likely that the model would probably need more context at later stages (when more sentences have already been seen), versus at the beginning of translating a long document.
3. In Figure 2, why does consistency drop so much for the en-xx 11-20 sentences bucket (and then increase again for longer sentence-wise distances) across all models?
4. The Workshop on Machine Translation (WMT, https://machinetranslate.org/wmt) also has document-level test sets, which are canonically used for MT evaluation in the literature. Why were evals presented on IWSLT2017 and Guofeng (relatively unknown datasets), but not on these standard test sets? Do the test sets used require long context to produce correct translations?

---

> ### Author Response · Authors · 2024-11-21
> **Response to Reviewer h7o9 (Chunk One)**
>
> > Q1. The Proper Noun Consistency metric is not sufficient to show the improvement of DelTA in terms of consistency compared to baselines.
>
> Proper noun translation consistency (also referred to as lexical or terminology translation consistency) is a well-recognized challenge in document-level translation[1,2]. For language pairs that do not share the same alphabet set, translating proper nouns across a document cannot simply rely on copying strings. The primary difficulty lies in maintaining a consistent translation for a proper noun throughout the entire document, which poses a significant challenge for LLMs with limited context capabilities.
> Moreover, as detailed in Section 6, we also conducted evaluations on pronoun and context-dependent translations, tasks that require more implicit contextual understanding to achieve accurate results. We believe these findings effectively demonstrate DelTA's improved ability to model discourse structures in documents during translation.
> In our future work, we plan to expand our study to more advanced consistency tasks in document-level translation, such as logical consistency. We appreciate your valuable feedback.
>
> [1] Lyu X, Li J, Gong Z, et al. Encouraging lexical translation consistency for document-level neural machine translation[C]//Proceedings of the 2021 Conference on Empirical Methods in Natural Language Processing. 2021: 3265-3277.
>
> [2] Karpinska M, Iyyer M. Large language models effectively leverage document-level context for literary translation, but critical errors persist[J]. arXiv preprint arXiv:2304.03245, 2023.
>
> > Q2. Significance tests are missing.
>
> The p-values of t-tests between DelTA and Sentence/Context methods are demonstrated in the following table:
>
> |                   | En-Xx |        | Xx-En |        |
> | ----------------- | ----- | ------ | ----- | ------ |
> | System            | COMET | LTCR-1 | COMET | LTCR-1 |
> | DelTA vs Sentence | <0.05 | 0.3472 | <0.05 | <0.05  |
> | DelTA vs Context  | <0.05 | 0.5867 | <0.05 | 0.2855 |
>
> Note that our framework achieves more consistency improvements on language pairs that do not share common alphabets, because proper nouns just need to be copied into the target sentences during translation for other language pairs. The p-values of LTCR-1 in En <=> Zh directions are less than 0.05.
>
> We have included these results in Section 5.2 of our paper. Thanks for your valuable feedback!
>
> > Q3. The performance gap between DelTA and context is small. If the Context model were enhanced with the Proper None Record module, would its performance on the proper noun consistency metric be improved?
>
> We conduct an extra comparative test between DelTA and the context method with Proper Noun Records attached. The results are shown in the Lines “DelTA” and “Short+Record” in the table in Q4. We can observe that there are fairly large performance gap between DelTA and the short+record method. This indicates that the different modules in the DelTA framework synergistically improve translation quality and consistency by extracting key information at multiple granularities and scales.

---

> ### Author Response · Authors · 2024-11-21
> **Response to Reviewer h7o9 (Chunk Two)**
>
> > Q4. The contribution of each auxiliary memory module is unclear.
>
> We apologize for the inadequacy of the ablation experiments due to space limitations. We report the ablation results for each module individually in the table below. The backbone model is GPT-3.5-Turbo, and the test set is IWSLT 2017 En-Zh (the same as the settings in the paper).
>
> | Modules                     | COMET | LTCR-1 |
> | --------------------------- | ----- | ------ |
> | None (Sentence)             | 83.78 | 80.27  |
> | Short-Term Memory (Context) | 84.50 | 77.89  |
> | Long-Term Memory            | 84.48 | 78.77  |
> | Long- & Short-Term Memory   | 84.54 | 79.23  |
> | Proper Noun Record          | 84.11 | 81.33  |
> | Summary                     | 84.51 | 79.73  |
> | Short+Record                | 84.45 | 82.37  |
> | All (DelTA)                 | 84.70 | 86.44  |
>
> We can observe that
> 1) Short-Term Memory enhances translation quality but negatively impacts consistency due to disruptions caused by the short-span context it introduces (Row "Short-Term Memory (Context)").
> 2) Similarly, Long-Term Memory improves translation quality but also leads to a decline in consistency (Row "Short-Term Memory"). However, these two modules complement each other, as their integration leverages information from different spans, resulting in further quality enhancements while mitigating consistency issues (Row "Long- & Short-Term Memory").
> 3) Proper Noun Records contribute to both translation quality and consistency, as improved accuracy in proper noun translation directly enhances overall quality (Row "Proper Noun Record").
> 4) Bilingual Summary also positively impacts translation quality by introducing the document's main idea, which guides the generation of target sentences more effectively (Row "Summary").
>
> Each module can bring its own performance improvement, but no single module can outperform the combination of them all. We believe this is because the architecture of DelTA can introduce auxiliary information from different granularities and scales to better improve translation quality and consistency, which is the synergy brought by our framework.
>
> We have added the more detailed ablation study results in Appendix E in our paper. Thanks for your opinion!
>
> > Q5. Why isn't the summary for chunk m conditioned on the summary for chunk m-1?
>
> In our implementation, we generate a summary for each chunk and iteratively merge it with the overall summary of the preceding text. This approach effectively conditions the new summary on its predecessor. However, we still conduct an additional experiment where the summary for chunk m is generated based on the summary of chunk m−1. Subsequently, this chunk-level summary is merged into the overall summary. The results of this experiment are presented in the following table:
>
> | System        | COMET | LTCR-1 |
> | ------------- | ----- | ------ |
> | Summary Cond. | 84.69 | 85.10  |
> | DelTA         | 84.70 | 86.44  |
>
> This approach brings no significant improvements, but we are considering generating the new overall summary directly on the condition of the old one in our future works, omitting the generation of chunk summaries. We appreciate your insightful suggestion.
>
> > Q6. The effect of adjusting the numbers of sentences to use as short- and long-term memory. The model would probably need more context at later stages?
>
> We conduct ablation studies on the window sizes of Short-Term and Long-Term Memory. The results are demonstrated in the following tables:
>
> | Short. Window | COMET | LTCR-1 |
> | ------------- | ----- | ------ |
> | 1             | 84.51 | 87.42  |
> | 3 (DelTA)     | 84.70 | 86.44  |
> | 5             | 84.65 | 87.42  |
>
> | Long. Window  | COMET | LTCR-1 |
> | ------------- | ----- | ------ |
> | 10            | 84.61 | 86.67  |
> | 20 (DelTA)    | 84.70 | 86.44  |
> | 30            | 84.71 | 84.25  |
> | Growing       | 84.68 | 85.57  |
>
> Expanding the Short-Term Memory window results in a slight improvement in consistency but significantly increases computational costs. This is because the translation prompt for each sentence is extended by two additional sentence pairs. Considering the trade-off between computational cost, translation quality, and consistency, we opted not to further increase the window size.
>
> Further increasing the Long-Term Memory window offers no significant benefits. Inspired by your suggestion that longer context windows might be more beneficial in the later stages of document translation, we experimented with a dynamic window size that grows by one sentence pair every eight sentences translated. However, this approach also failed to deliver performance improvements. We believe that the key information from more distant contexts is effectively captured by the Summary module. The iterative process of generating and merging summaries can itself be viewed as an alternative mechanism for expanding the context window.
>
> This part of the experiments has also been added to Appendix E in our paper.

---

> ### Author Response · Authors · 2024-11-21
> **Response to Reviewer h7o9 (Chunk Three)**
>
> > Q7. Why does consistency drop significantly for the En-Xx 11-20 sentence distance bucket?
>
> We believe this phenomenon is attributed to the inherent distribution of the data, as all systems show a drop in consistency rates within this bucket, while such behavior is not observed in other translation directions.
>
> > Q8. Why use IWSLT2017 and Guofeng instead of standard WMT test sets?
>
> There is a misunderstanding. The Guofeng dataset is indeed the official data set of the WMT23 literary translation task which is mainly composed of web novels, and we conduct our experiments on the Test 2 dataset  (see https://www2.statmt.org/wmt23/literary-translation-task.html for more details). IWSLT2017 is another widely used document-level translation dataset, characterizing long TED speeches. We believe that these datasets with long documents are challenging benchmarks for document-level translation, which our DelTA framework is still capable of.

---

> > ### Comment · Reviewer_h7o9 · 2024-11-22
> >
> > Thank you for your substantial additional results, which addressed my major concerns. I have raised my soundness and overall scores.

---

### Official Review · Reviewer_oGf7 · 2024-11-03

**Soundness:** 3
**Presentation:** 3
**Contribution:** 2
**Rating:** 6
**Confidence:** 4

**Summary:**

The paper proposes DelTA, a document-level translation agent that can translate documents or long context text with better translation quality than other methods. The approach consists of a series of prompts that are used to encode information about proper nouns, immediate context, larger context, summaries in the source and target language. Experiments are carried out in the IWSLT2017 benchmark for 4 language pairs and evaluated for consistency of translation terms and translation quality using reference-based translation evaluation methods.

**Strengths:**

The idea of the paper is simple and straightforward: provide more context about the translation task by presenting this information in the prompt of an LLM. The context is varied: sentences occurring in the immediate before the sentence being translated, context that is longer with sentences much before the current sentence, a summary of the source text, a summary of the target text and proper nouns occurring in the text with their respective translations.

**Weaknesses:**

The evaluation was performed only with automatic metrics such as sentence-level and document-level COMET. It would be interesting to understand if the performance reported holds with evaluation performed by translators in terms of quality and consistency as both implementations of COMET are known to be weak proxies for translation quality beyond the sentence-level, failing to capture consistency, coherence and other discourse-related linguistic dimensions.

**Questions:**

* Consider using the WMT24 shared task benchmark test sets that are going to be released. They provide document-level test sets. In 2023 the organizers released test sets only for English-German but for this year more language pairs will be made available.

---

> ### Author Response · Authors · 2024-11-21
> **Response to Reviewer oGf7**
>
> > Q1. The involved automatic metrics fail to capture discourse-related linguistic dimensions.
>
>  It is widely acknowledged that evaluating document-level translation using a unified automated metric poses significant challenges. Meanwhile, as outlined in Section 6, we conducted additional evaluations focusing on pronoun and context-dependent translation accuracy, tasks that require a deeper understanding of implicit contextual information to produce coherent results. We hope these findings highlight DelTA's enhanced capability to model discourse structures in documents to some extent. Due to time constraints, we will consider adding human preference evaluations in our paper's revised version. Thanks for your insightful opinion!
>
> > Q2. Consider using the WMT24 shared task benchmark test sets.
>
> Thank you for your valuable advice! We have already evaluated our system on Guofeng, the official test set for literary documents in WMT23, to demonstrate our progress. We look forward to further testing DelTA on additional language pairs and enhancing our framework based on these test sets as they are released.

---

> > ### Comment · Reviewer_oGf7 · 2024-11-21
> >
> > Thank you for the clarifications.

---

> > > ### Author Response · Authors · 2024-11-21
> > >
> > > We’re glad we could address your concerns. Please don’t hesitate to let us know if you have any further questions or suggestions. We would be happy to discuss them.

---

### Official Review · Reviewer_bC5N · 2024-11-06

**Soundness:** 3
**Presentation:** 4
**Contribution:** 3
**Rating:** 8
**Confidence:** 4

**Summary:**

This paper presents a methodology that improves translation consistency by using an agent-based approach. DelTA utilizes a modular, multi-level memory system, comprising Proper Noun Records, Bilingual Summary, Long-Term Memory, and Short-Term Memory, which store and manage information at different scales, with auxiliary LLM-driven components ensuring efficient information retrieval and updates.

**Strengths:**

* This paper effectively addresses a significant challenge in the field, presenting meaningful improvements and contributions.

*The experimental design is comprehensive and well-executed, providing robust results.

*The presentation is clear, concise, and well-organized, facilitating easy understanding of the research.

**Weaknesses:**

* As authors point out in limitations, efficiency of the method

**Questions:**

* How well does this scale to extremely low-resource languages?

---

> ### Author Response · Authors · 2024-11-21
> **Response to Reviewer bC5N**
>
> > Q1. As authors point out in limitations, the efficiency of the method is relatively low.
>
> To overcome these limitations, we propose several potential improvements. First, introducing special boundary tags to explicitly mark sentence boundaries can enable sentence-level alignment within generated paragraphs, allowing LLMs to process multiple sentences simultaneously and reduce invocation frequency. Second, using more precise alignment tools and scripts to extract proper nouns and their translations, instead of relying on LLMs, can improve the efficiency of DelTA. Additionally, components like the Long-Term Retriever could be implemented with a dense retriever rather than depending on LLMs to identify related sentences. In the summary component, streamlining the process by merging sentences within the window directly into a single-step summary can further reduce runtime. Last but not least, training an inference trajectory designer to decide, on a case-by-case basis, whether a specific component should be invoked--a focus of recent research on intelligent agents--can add an additional layer of efficiency. These proposed optimizations will serve as key areas for future work.
>
> > Q2. How well does this scale to extremely low-resource languages?
>
> We evaluate how well DelTA scales to low-resource languages.
> Considering that the spaCy NLP tool used for evaluation does not support many low-resource languages, we select Lithuanian => English as the language pair to test.
> We randomly sample a document (with 556 sentences) from the Europarl v9 training set as our evaluation set and employ GPT-3.5-Turbo as the backbone model.
> The results are shown in the following table, indicating that DelTA also performs well on the low-resource language pair:
>
> | System   | COMET | LTCR-1 |
> | -------- | ----- | ------ |
> | Sentence | 87.33 | 78.65  |
> | Context  | 87.87 | 78.09  |
> | DelTA    | 87.96 | 82.68  |
>
> We have added this part in Appendix F in our paper. Thanks for your advice!

---

### Comment · Area_Chair_wxqk · 2024-11-21
**Reminder: Please respond and update the score if necessary**

Dear Reviewers,

Kindly ensure that you respond proactively to the authors' replies (once they are available) so we can foster a productive discussion. If necessary, please update your score accordingly. We greatly appreciate the time and effort you’ve dedicated to the review process, and your contributions are key to making this process run smoothly.

Thank you,

AC

---

### Author Response · Authors · 2024-11-21
**Response to All Reviewers**

We greatly appreciate your insightful suggestions and opinions. Our paper has been modified and updated based on your feedback, with changes highlighted in cyan (for example, in the abstract). Additionally, we have released our code and data in an  ANONYMOUS repository: https://anonymous.4open.science/r/DelTA_Agent-7716. We look forward to further discussions!

---

### Meta-Review · Area_Chair_wxqk · 2024-12-22

**Metareview:**

This paper introduces DelTA, a Document-level Translation Agent designed to enhance translation consistency and quality through an agent-based, multi-level memory system. DelTA uses four memory components: Proper Noun Records, Bilingual Summary, Long-Term Memory, and Short-Term Memory, to effectively store and retrieve information at varying levels of detail. By utilizing LLM-driven components, DelTA ensures efficient information management and retrieval. The methodology is tested on the IWSLT2017 benchmark across four language pairs, showing superior translation quality compared to traditional methods. Evaluation employs both standard and novel consistency metrics, demonstrating notable improvements in translation consistency and quality.

On the positive side, the paper offers a straightforward yet impactful approach to enhancing translation quality by incorporating additional context into the prompts given to large language models (LLMs). This context includes the immediate preceding sentences, longer contexts from earlier in the text, summaries of both the source and target texts, and proper nouns with their translations. The motivation for this research is well-articulated, highlighting the significance and relative underexploration of document-level machine translation (MT) in the current landscape. DelTA, the system proposed, is notably easy to implement as it primarily involves prompting existing LLMs. Each module within the DelTA system is clearly justified, contributing to the paper's clear and accessible presentation of its experimental setup. The results are thorough, covering a variety of LLMs and language pairs, and include comprehensive comparisons with key document-level MT baselines.

Areas for improvement, as highlighted by the reviewer, include the reliance on automatic metrics such as sentence-level and document-level COMET for evaluation. To gain a more comprehensive understanding of the system's performance, it would be beneficial to incorporate assessments by human translators, particularly regarding quality and consistency. Current COMET implementations are recognized as weak proxies for translation quality beyond the sentence level, often failing to adequately capture elements such as consistency, coherence, and other discourse-related linguistic aspects. Additionally, the initial evaluations lacked significance testing, a concern that the authors have since addressed and updated in the revised paper. Another concern I have regarding this paper is its narrow focus on document translation, a specific aspect of natural language processing (NLP). While the findings may be valuable to a limited segment of the scientific community, they may not have broad applicability across the field of NLP as a whole, including ICLR audiences.  Consequently, while the paper presents significant merits within the machine translation field, I am learning to recommend acceptance.

Its impact appears to be more confined to a specialized audience rather than offering broad applicability across the wider NLP community. As such, it may be more advantageous for this paper to be presented at an NLP-focused conference where its specialized contributions would be better appreciated.

**Additional Comments On Reviewer Discussion:**

Reviewer oGf7 mentions the simple and effective idea of providing contextual information in LLM prompts for better translation. However, they note the reliance on automatic metrics like COMET, which may not comprehensively assess translation quality and recommend incorporating human evaluations for a more robust analysis.

Reviewer bC5N acknowledges the paper's significant contributions to addressing key challenges in document translation, with a clear presentation and comprehensive experimental design. However, they express concerns over the method's efficiency.

Reviewer gPC6 highlights the originality and quality of the multi-level memory architecture and its novel approach to document translation, as well as the introduction of metrics for evaluating translation consistency. They point out weaknesses such as the need for more detailed efficiency analysis, missing comparisons with relevant baselines, incomplete motivation and analysis of context-dependent translation, and a lack of technical details and plans for code release. They also emphasize the importance of statistical significance testing and referencing related literature to support claims.

Reviewer h7o9 commends the paper for addressing document-level translation, a vital yet underexplored area in machine translation. The Delta system is noted for its ease of implementation using off-the-shelf LLMs, and the paper's experimental setup is thorough, comparing multiple LLMs and language pairs against key baselines. However, the paper primarily measures Proper Noun Inconsistency, not addressing more complex inconsistency challenges, which weakens some claims about its effectiveness. Comparisons show "Context" often outperforming Delta, but without significance testing, the evidence is less convincing. Additionally, the paper lacks detailed analysis of the contributions of Delta's five auxiliary modules.

---

### Decision · Program_Chairs · 2025-01-22

Accept (Poster)